# Bacterial outer membrane vesicle based versatile nanosystem boosts the efferocytosis blockade triggered tumor-specific immunity

Wan-Ru Zhuang[1], Yunfeng Wang[1], Weidong Nie[1], Yao Lei[1], Chao Liang[1], Jiaqi He[1], Liping Zuo[1], Li-Li Huang[2] & Hai-Yan Xie [1] ✉

Efferocytosis inhibition is emerging as an attractive strategy for antitumor immune therapy because of the subsequent leak of abundant immunogenic contents. However, the practical efficacy is seriously impeded by the immunosuppressive tumor microenvironments. Here, we construct a versatile nanosystem that can not only inhibit the efferocytosis but also boost the following antitumor immunity. MerTK inhibitor UNC2025 is loaded into the bacterial outer membrane vesicles (OMVs), which are then modified with maleimide (mU@OMVs). The prepared mU@OMVs effectively inhibits the efferocytosis by promoting the uptake while preventing the MerTK phosphorylation of tumor associated macrophages, and then captures the released antigens through forming universal thioether bonds. The obtained in situ vaccine effectively transfers to lymph nodes by virtue of the intrinsic features of OMVs, and then provokes intense immune responses that can efficiently prevent the growth, metastasis and recurrence of tumors in mice, providing a generalizable strategy for cancer immunotherapy.

Efferocytosis, a phagocytic process to remove dying apoptotic cells, is crucial for normal cell development and tissue homeostasis[1–3]. In tumors, widespread cellular stress from uncontrolled proliferation generates large amounts of apoptotic cells, which also lead to ubiquitous efferocytosis by tumor-associated macrophages (TAMs)[4,5]. Consequently, the efferocytosis-related signaling pathways are triggered, resulting in the reduction of proinflammatory cytokines, inhibition of M1-macrophages polarization, restriction of antigen presentation of macrophages, increase of Treg cells while decrease of CD8[+] T cells in tumor microenvironment (TME), further promoting tumor proliferation, invasion, and metastasis[6,7]. Therefore, efferocytosis is closely linked to tumor development, and efferocytosis inhibition is emerging as a novel strategy for antineoplastic treatment[8]. In macrophages, the myeloid-epithelial-reproductive tyrosine kinase (MerTK) is the crucial phagocytic receptor that recognizes the externalized "eat-me" signal phosphatidylserine (PtdSer) on apoptotic cells[9,10]. Hence, MerTK is an effective target for the efferocytosis

blockade. Both MerTK inhibitors and anti-MerTK antibodies have been used to selectively avoid the phosphorylation of MerTK and then impede the efferocytosis by TAMs, resulting in delayed clearance of apoptotic cells[11–14]. Following with the secondary necrosis of tumor cells, a variety of immunogenic contents, including damage-associated molecular patterns (DAMPs) and tumor-associated antigens (TAAs), are released to the TME and initiate diverse antitumor immune responses[15,16]. Especially, the entire spectrum of TAAs released from tumor cells is favorable for the construction of in situ therapeutic cancer vaccine, which is obviously superior to the conventional vaccines in fighting against heterogeneous tumors[17,18].

Although effective, the treatment effects of taking advantage of in situ antigens are inevitably disturbed by the immunosuppressive TME. First of all, dendritic cells (DCs), essential for antigen recognition and presentation, are typically dysfunctional and rare in TME[19,20]. Moreover, antigens cannot cross through the tumor stroma and may be quickly cleared by the tumor tissues[21]. Recruiting more DCs and

[1]School of Life Science, Beijing Institute of Technology, 100081 Beijing, P.R. China. [2]School of Medical Technology, Beijing Institute of Technology, 100081 Beijing, P.R. China. ✉e-mail: hyanxie@bit.edu.cn

restoring their function by immune factors or activators can partly solve the above problems, whereas the immunosuppressive TME is difficult to reverse and will cause the dysfunction of DCs again[22,23]. As an alternative, transferring the in situ TAAs out of tumors, especially to immune cell-enriched lymph nodes (LNs), is more effective in activating the anti-tumoral immune response, but hard to realize[24]. In addition, antigens alone are usually unable to elicit potent immune responses, and an efficient tumor vaccine usually contains immunostimulatory adjuvants[25]. As one meaningful attempt, a gel system was constructed to transfer antigens to LNs by cooperating with multiple capacities, including phototherapy and chemotherapy-induced antigen release, antigen adsorption, LNs targeting as well as adjuvant supply[26]. However, tumors often develop resistance to such therapies and apoptotic tumor cells still face the risk of efferocytosis, reducing the release of TAAs and then weakening the immunologic effects[27]. Moreover, such a combining concept is difficult to be applied generally. An efficient strategy that can not only prompt the robust release of TAAs in situ but also the transfer of in situ TAAs to LNs is still in great expectation.

Herein, we fabricate a versatile nanosystem for efficient tumor immunotherapy by synergizing the efferocytosis blockade-induced secondary necrosis with the bacterial outer membrane vesicles (OMVs)-based antigen transfer and immune augment. One of the small-molecule MerTK inhibitors UNC2025 is loaded into OMVs, which is then modified with maleimide (Mal) (Fig. 1a). After peritumoral injection, the prepared mU@OMVs is easily recognized and phagocytized by TAMs, resulting in efficient inhibition of the phosphorylation of MerTK in TAMs, and then the efferocytosis blockade of apoptotic cells and the secondary necrosis of tumor cells. The consequently in situ released TAAs are then captured by mU@OMVs through forming stable thioether bonds with Mal and efficiently hitchhiked to LNs with the aid of OMVs. The codelivery of massive TAAs and novel adjuvant OMVs to LNs provoke effective antigen presentation and DCs maturation, and then intense CD8+ T-based immunotherapeutic efficacy in the xenografted, metastatic as well as recurrent tumor models of mice, providing a potential strategy for putting forward the cancer immunotherapy.

## Results
### Preparation and characterization of mU@OMVs
OMVs were extracted from the *Escherichia coli* MG1655 as reported before[28]. Afterward, the hydrophilic UNC2025 was encapsulated into OMVs through electroporation[29]. The loading saturated when the mass ratio of UNC2025 and OMVs was 1:1. The corresponding loading efficiency (LE) was 21.72% and the encapsulation efficiency (EE) was 27.76% (Supplementary Fig. 1a, b). Mal was then decorated onto the surface of OMVs through a reaction between Mal-PEG4-NHS and the amines on OMVs[30], a process with negligible influence on UNC2025 loading and encapsulation (Fig. 1b and Supplementary Fig. 2). The UNC2025 loading and Mal modified OMVs (mU@OMVs) exhibited the characteristic absorption peaks of UNC2025 in the UV-vis absorbance spectra (Fig. 1c). Transmission electronic microscopy (TEM) imaging further confirmed that both UNC2025 encapsulated OMVs (U@OMVs) and mU@OMVs displayed a typical vesicular morphology similar to that of OMVs (Fig. 1d and Supplementary Fig. 3). Also, OMVs, U@OMVs as well as mU@OMVs displayed a consistent protein distribution (Supplementary Fig. 4). Dynamic laser scattering (DLS) and nanoparticle tracking analysis (NTA) revealed the uniform size distributions and satisfactory size stability of all the three vesicles (Fig. 1e, g and Supplementary Fig. 5). The DLS sizes of OMVs, U@OMVs and mU@OMVs were steadily increased from $38.69 \pm 4.84$ to $77.09 \pm 14.09$ and $100.04 \pm 1.70$ nm (Fig. 1f). Moreover, the particle concentrations of U@OMVs and mU@OMVs were maintained at nearly 90% compared with that of OMVs (Supplementary Table 1). Mal-modified OMVs have been approved to be able to effectively capture the tumor-released

proteins[30]. To explore the antigen capture capacity of our mU@OMVs, different formations of OMVs were incubated with the model antigen ovalbumin (OVA) in vitro. As could be seen, mU@OMVs showed almost 2-fold higher OVA adsorption level than that of OMVs or U@OMVs (Fig. 1h, i), illustrating the well-kept chemical reaction ability of Mal on OMVs surface, and thus mU@OMVs could capture antigens with high efficiency.

### mU@OMVs-induced efferocytosis blockade for TAAs liberation
In immunosuppressive tumors, M2-like TAMs make up the majority of macrophages so we used M2 macrophages here to investigate the mU@OMVs-induced efferocytosis blockade[31]. First, the M0 macrophages were polarized to the M2 phenotype by interleukin-4 (IL-4) (Supplementary Fig. 6). Then, mU@OMVs could be effectively uptaken by M2 macrophages as native OMVs owing to the multiple immunogenic components derived from parental bacteria[31], while did not compromise the cell viability of macrophages (Supplementary Figs. 7, 8). UNC2025 is a typical inhibitor that can effectively restrain the kinase activity of MerTK by preventing its phosphorylation[32,33]. Soon after internalization, mU@OMVs was able to inhibit the phosphorylation of MerTK (p-MerTK) in a dose-dependent manner similar to that of free UNC2025 without affecting other PtdSer receptors like Axl (Fig. 2a and Supplementary Figs. 9–11). The decrease of p-MerTK would interfere with the activation of MerTK signaling and then induce the efferocytosis suppression (Fig. 2b). For verification, we next investigated the phagocytosis of apoptotic tumor cells by MerTK-inhibited M2 phenotype macrophages, which are the most representative macrophage population in tumors. The model apoptotic tumor cells (AC) were obtained by adding doxorubicin to B16F10 melanoma cells for 6 h, and around 60% of B16F10 cells exposed PtdSer on the surface (Supplementary Fig. 12). M2 macrophages were pre-incubated-with-free UNC2025 or different formations of OMVs, and then collected to incubate with the apoptotic tumor cells. As expected, M2 macrophages without treatment could hardly phagocytize living tumor cells (M-LC group) but cleared about 80% of apoptotic tumor cells (M-AC group), and OMVs treatment (M-AC + OMVs group) cannot alleviate this clearance because strong PtdSer signaling bridging to MerTK would induce active efferocytosis (Fig. 2c–e). Nevertheless, UNC2025, U@OMVs, or mU@OMVs pre-treated M2 macrophages (M-AC + UNC2025, M-AC + U@OMVs, or M-AC + mU@OMVs groups) distinctly decreased the phagocytosis of apoptotic cells to ~10%, and this decrease could be sustained for 18 h after co-incubation (Supplementary Fig. 13), indicating the significant efferocytosis blockade capacity of mU@OMVs due to UNC2025-induced MerTK inhibition.

The efficient efferocytosis suppression would lead to the subsequent secondary necrosis of apoptotic cells. Nuclear high mobility group 1 (HMGB1) proteins, also known as DAMPs, are the main signals of necrosis since they cannot be secreted by apoptotic cells but are passively leaked from necrotic cells owing to the breakage of the membrane integrity (Fig. 2b)[34]. To verify the secondary necrosis of the tumor cells, we detected the level of HMGB1 in the supernatant of the above groups by enzyme-linked immunosorbent assays (ELISA). As shown in Fig. 2f, the release of HMGB1 from the apoptotic tumor cells (AC group) was obvious but faint in M-LC group. The co-incubation of M2 macrophages with apoptotic tumor cells (M-AC group) significantly decreased the HMGB1 level owing to the efferocytosis effect of M2 macrophages, and the addition of OMVs did not ameliorate the HMGB1 liberation due to the feeble efferocytosis inhibition ability of OMVs alone. However, the release of HMGB1 was remarkably promoted in M-AC + UNC2025, M-AC + U@OMVs, and M-AC + mU@OMVs groups. Typically, the supernatant HMGB1 level was only 1 ng/mL in M-AC group but increased by almost two folds in M-AC + mU@OMVs group, emphasizing the widespread secondary necrosis of apoptotic tumor cells attributed to the highly effective efferocytosis inhibition.

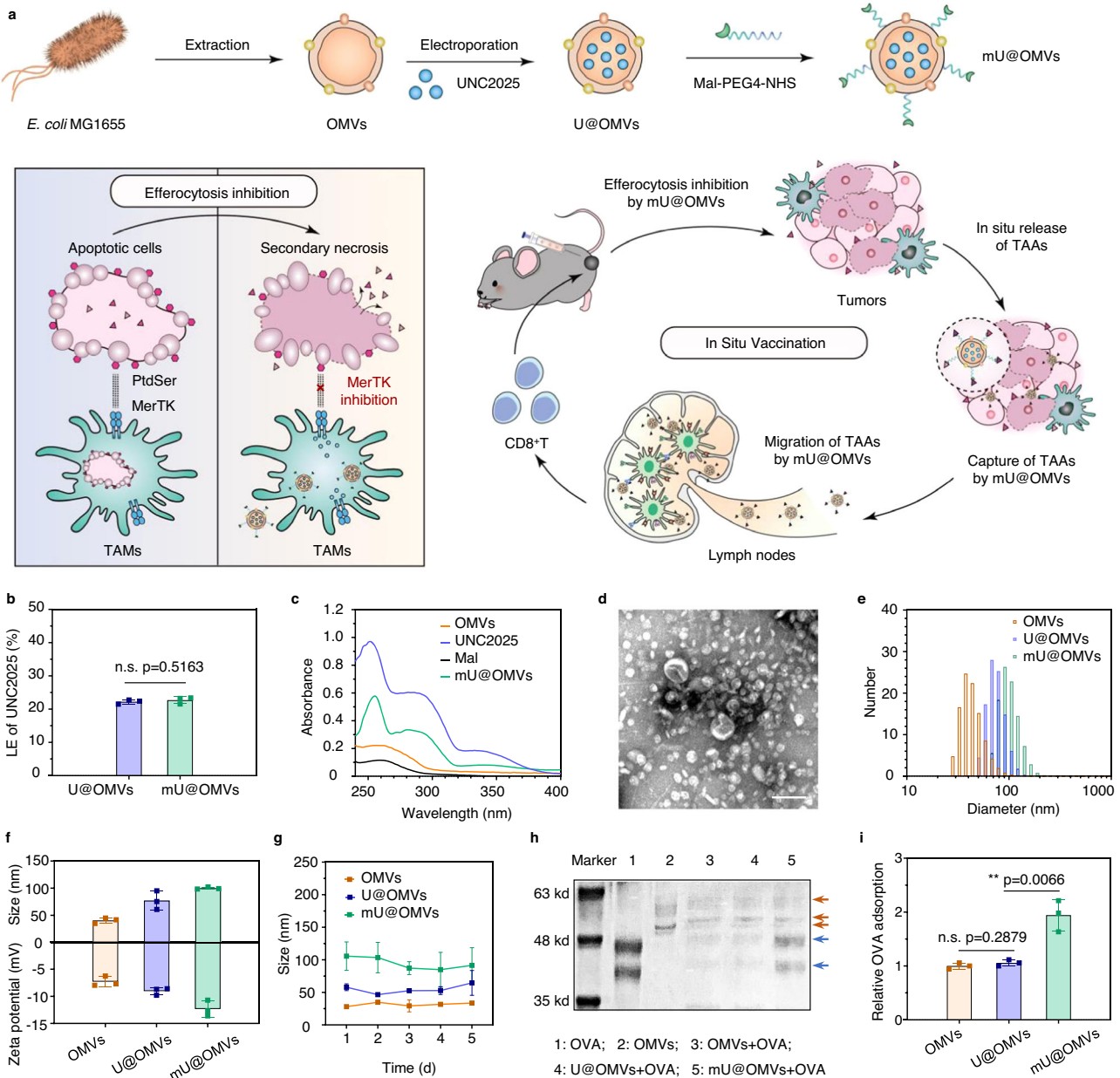

**Fig. 1 | Schematic and characterization of mU@OMVs. a** Preparation and therapeutic strategy of mU@OMVs for in situ cancer vaccination. **b** Loading efficiency (LE) of UNC2025 before and after maleimide (Mal) modification. **c** UV-Vis absorbance spectra. **d** TEM image of mU@OMVs. Scale bar: 100 nm. **e** Size distribution. **f** Size and zeta potential. **g** Stability of mU@OMVs in PBS at 4 °C. **h** SDS-PAGE of OMVs, U@OMVs, and mU@OMVs before and after incubation with ovalbumin (OVA) (yellow arrows: proteins bands of OMVs; blue arrows: proteins bands of OVA). **i** Relative adsorption of OVA by OMVs, U@OMVs, and mU@OMVs determined by the bicinchoninic acid (BCA) assay. Data in **b**, **f**, **g**, **i** are presented as mean ± s.d. ($n = 3$ biologically independent samples). Statistically significant differences between groups were identified by unpaired two-tailed Student's $t$-test. ****$P < 0.0001$, ***$P < 0.001$, **$P < 0.01$, *$P < 0.05$, n.s., not significant. Source data are provided as a Source Data file.

The released contents would then encounter and conjugate with mU@OMVs. For verification, mU@OMVs were incubated with the supernatant of M-AC + mU@OMVs for 3 h and then collected for proteomics analysis. As could be seen, mU@OMVs captured not only DAMPs but also massive tumor mutational proteins attributed to the universal maleimide-thiol reactions between Mal on mU@OMVs and thiol moieties in tumor-associated proteins (Fig. 2g and Supplementary Table 2)[35]. Especially, tumor neoantigens as well as some predicted neoantigens, which are powerful for tumor-specific immune activation, were also captured by mU@OMVs (Fig. 2h). All these results together clarified that mU@OMVs not only efficiently inhibited the efferocytosis of apoptotic cells but also captured a great diversity of DAMPs and TAAs released from the necrotic cells.

## In vitro immune cells activation by antigen-captured mU@OMVs

The efficient antigen capture by mU@OMVs together with the well-known adjuvanticity of OMVs implied the incredible potential of mU@OMVs to boost potent antitumor immune responses. The antigen presentation by DCs is the first step of immune response initiated from antigen internalization[36]. OMVs can be efficiently uptaken by DCs owing to the strong recognition between PAMPs on OMVs and PRRs on DCs[37,38]. Hence, mU@OMVs would promote the uptake of captured antigens by DCs. For visible verification, mU@OMVs were mixed with PE-labeled OVA at first and then incubated with bone marrow-derived DCs (BMDCs) overnight. As could be seen, considerable OVA were internalized by BMDCs in OVA + mU@OMVs group by comparison

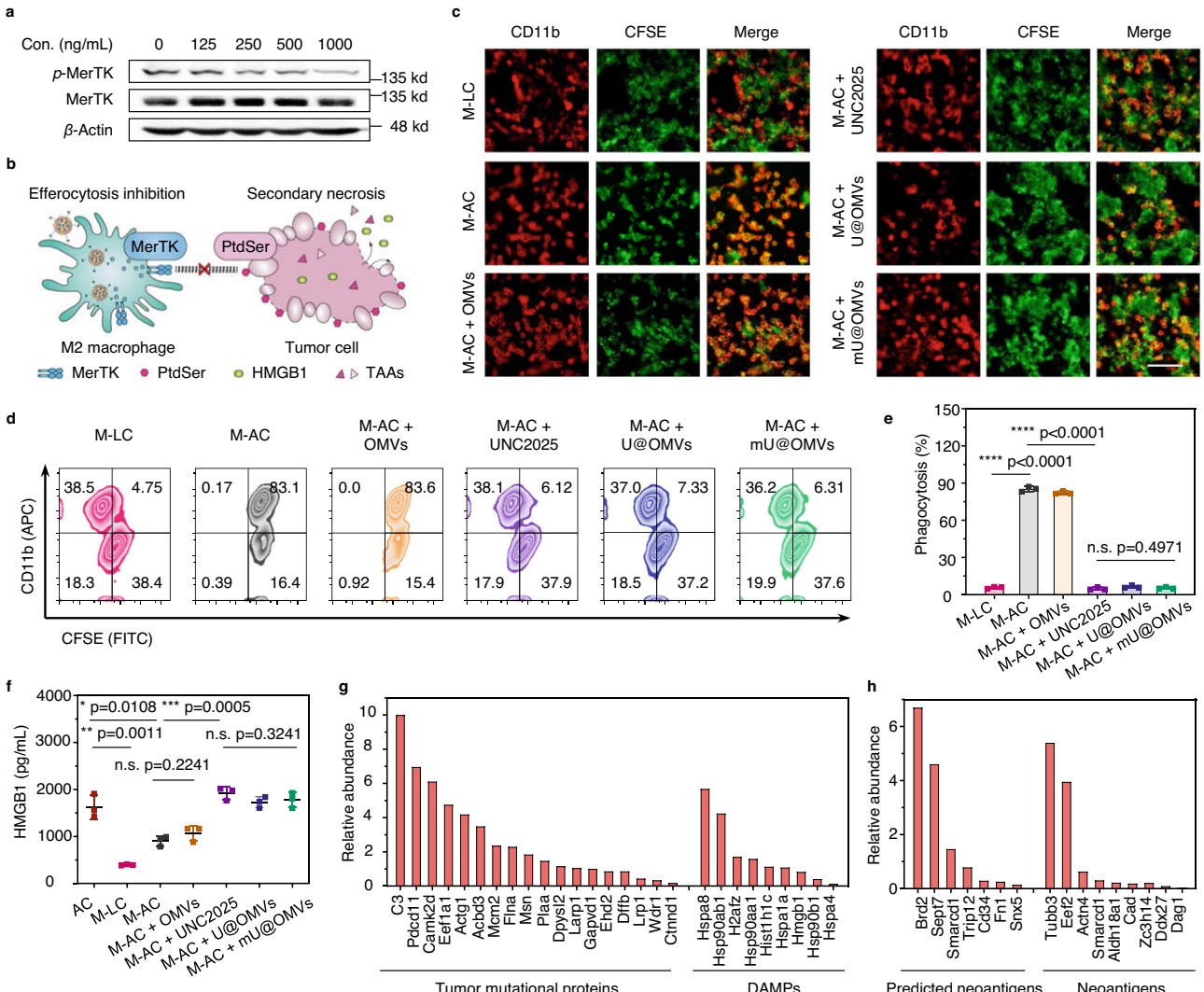

**Fig. 2 | Efferocytosis blockade by mU@OMVs for the release of TAAs. a** Western blot analysis of phosphorylation of MerTK (p-MerTK) in M2 macrophages after mU@OMVs treatment for 2 h. **b** Schematic of mU@OMVs-mediated efferocytosis inhibition and antigen release. **c** CLMS imaging of the engulfment of AC (carboxyfluorescein succinimidyl ester (CFSE)-labeled, green) by macrophages (APC-conjugated anti-CD11b labeled, red) after a 6-h incubation. M: M2 macrophages; LC: living B16F10 cells; AC: apoptotic B16F10 cells. Scale bar: 50 μm. **a**, **c** Data are representative of three biological replicates. **d**, **e** Flow cytometry and quantification of phagocytosis after different treatments. **f** HMGB1 levels in the supernatants were determined by ELISA after further 18 h culture. **g**, **h** Relative abundance of tumor mutational proteins, DAMPs, predicted neoantigens and neoantigens captured by mU@OMVs. Proteins-adsorbed mU@OMVs were collected for proteomics analysis and identified according to the previous report[21,46,47]. Data in **e**, **f** are presented as mean ± s.d. ($n = 3$ biologically independent samples). Statistically significant differences between groups were identified by unpaired two-tailed Student's $t$-test. ****$P < 0.0001$, ***$P < 0.001$, **$P < 0.01$, *$P < 0.05$, n.s., not significant. Source data are provided as a Source Data file.

with free OVA (Fig. 3a, b and Supplementary Fig. 14), and the mean fluorescence intensity (MFI) increased more than 2 folds (Fig. 3c), demonstrating that mU@OMVs significantly enhanced the uptake of antigens by DCs.

We then evaluated whether the enhanced delivery of antigens could promote the maturation of DCs as well as the activation of cytotoxic T cells. TAAs obtained from the supernatant of necrotic B16F10 cells were preincubated with OMVs (TAAs + OMVs group), U@OMVs (TAAs + U@OMVs group), or mU@OMVs (TAAs + mU@OMVs group), and then these different formations, free TAAs, and OMVs were individually added into BMDCs and incubated for 24 h (Fig. 3d). We found that TAAs alone showed inferior ability in inducing DC maturation, probably due to its limited internalization into DCs. OMVs significantly upregulated the expression of typical markers of mature DCs, such as CD80, CD86, MHC-II, and MHC-I (Fig. 3e–h and Supplementary Fig. 15), associated with the fact that the adjuvanticity

of PAMPs-abundant OMVs can induce the maturation of DCs[31]. The expressions of these markers were not further enhanced in TAAs + OMVs or TAAs + U@OMVs group, probably owing to the weak interaction between TAAs with OMVs or U@OMVs and thus ineffective ingestion of TAAs. Nevertheless, all the proportions of CD80[+], CD86[+], MHC-II[+], or MHC-I[+] DCs in the TAAs + mU@OMVs group were increased by more than 10% compared with that of OMVs group. Additionally, TAAs + mU@OMVs-treated BMDCs secreted much more interleukin-6 (IL-6), tumor necrosis factor-α (TNF-α), and interleukin-12p40 (IL-12p40) than other groups (Supplementary Fig. 16). These results verified that TAAs + mU@OMVs treatment significantly promoted the DC maturation, benefiting from the enhanced antigen untaken as well as the coexistence of TAAs and adjuvant OMVs. This would then facilitate antigen presentation and T-cell activation. Accordingly, the percentage of CD69[+] T cells in the TAAs + mU@OMVs group was the highest and increased by 14.8% and 16.1% to that of free

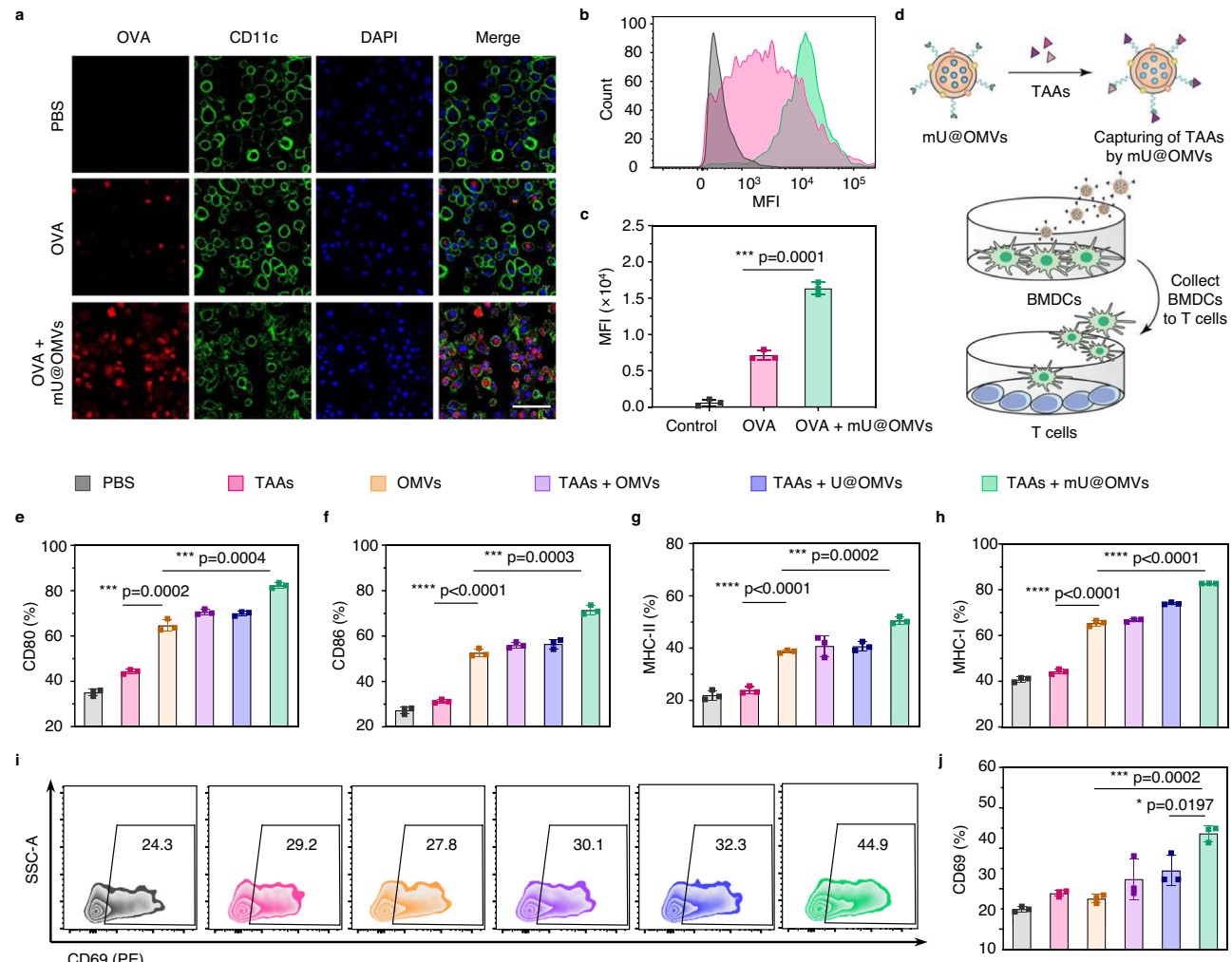

**Fig. 3 | Antigen uptake and immune cell activation by mU@OMVs. a** CLMS images of BMDCs with PBS, OVA, or OVA + mU@OMVs treatment for 12 h. OVA (PE-labeled, red); BMDCs (FITC-conjugated anti-CD11c labeled, green); cell nucleus (Hoechst 33342, blue). Scale bar: 50 μm. **b**, **c** Flow cytometry and quantification of the internalization after different treatments. **d** Schematic illustration of the experiment of immune cell activation. TAAs were incubated with different vesicles and added into BMDCs for 24 h. Then, parts of BMDCs were collected and cultured with T cells for a further 24 h. The cells and the supernatant from different treatments were collected for flow cytometry assay or ELISA. **e–h** Expression levels of CD80, CD86, MHC-II, and MHC-I on BMDCs by flow cytometry. **i**, **j** Flow cytometry and quantification of expression levels of CD69 gated on CD8+ T cells. Data in **c**, **e–h**, **j** are presented as mean ± s.d. ($n = 3$ biologically independent cells). Statistically significant differences between groups were identified by unpaired two-tailed Student's t-test. ****$P < 0.0001$, ***$P < 0.001$, **$P < 0.01$, *$P < 0.05$. Source data are provided as a Source Data file.

TAAs and OMVs, respectively (Fig. 3i, j and Supplementary Fig. 17), illustrating the high efficient activation of T cells. In short, our results confirmed that the antigen-captured mU@OMVs could induce a vigorous immune response in vitro attributed to the codelivery of abundant TAAs and powerful adjuvant OMVs into DCs.

### In vivo efferocytosis blockade and antigen transfer to LNs by mU@OMVs

Then, we were in the position to explore the performance of mU@OMVs in vivo. First of all, to evaluate the efferocytosis suppression effects of mU@OMVs, B16F10 melanoma-bearing mice were peritumorally treated with PBS, UNC2025, OMVs, U@OMVs, and mU@OMVs, respectively. 48 h after injection, tumor sections were collected and stained for the terminal deoxynucleotidyl transferase-mediated deoxyuridine triphosphate nick end labeling (TUNEL) and cleaved-Caspase 3 (c-Casp3) immunohistochemistry assays. Consistent with the in vitro efferocytosis inhibition results, the percentage of TUNEL+ or c-Casp3+ cells in the UNC2025 group significantly increased compared to that of PBS control owing to the enhanced

accumulation of apoptotic cells in tumors after the MerTK inhibition (Fig. 4a–d). It was worth noting that OMVs treatment also invoked certain apoptosis, probably due to the OMVs induced IFN-γ increase[28,39]. As expected, both U@OMVs and mU@OMVs potently increased the proportions of TUNEL+ or c-Casp3+ cells than that of free UNC2025 or OMVs group owing to the cooperation of OMVs and UNC2025. To confirm the subsequent release of intracellular contents, the HMGB1 level in the serum of tumor-bearing mice after different treatments were analyzed by ELISA (Fig. 4e). Compared with PBS, UNC2025 treatment increased the level of HMGB1 in blood, while the effect of OMVs was slight. This was because although OMVs could cause the apoptosis of tumor cells to a certain extent, TAMs in tumors might restrain the following necrosis progress by rapid efferocytosis. In contrast, the apoptotic cells with attenuated activity of MerTK successively developed into secondary necrosis and then HMGB1 was released. Notably, the levels of HMGB1 in U@OMVs or mU@OMVs treated mice were further improved by approximately 1.2 and 1.5 times than that of the UNC2025 group, again illustrating the smart synergism of UNC2025 and OMVs in inducing necrosis, and then the robust

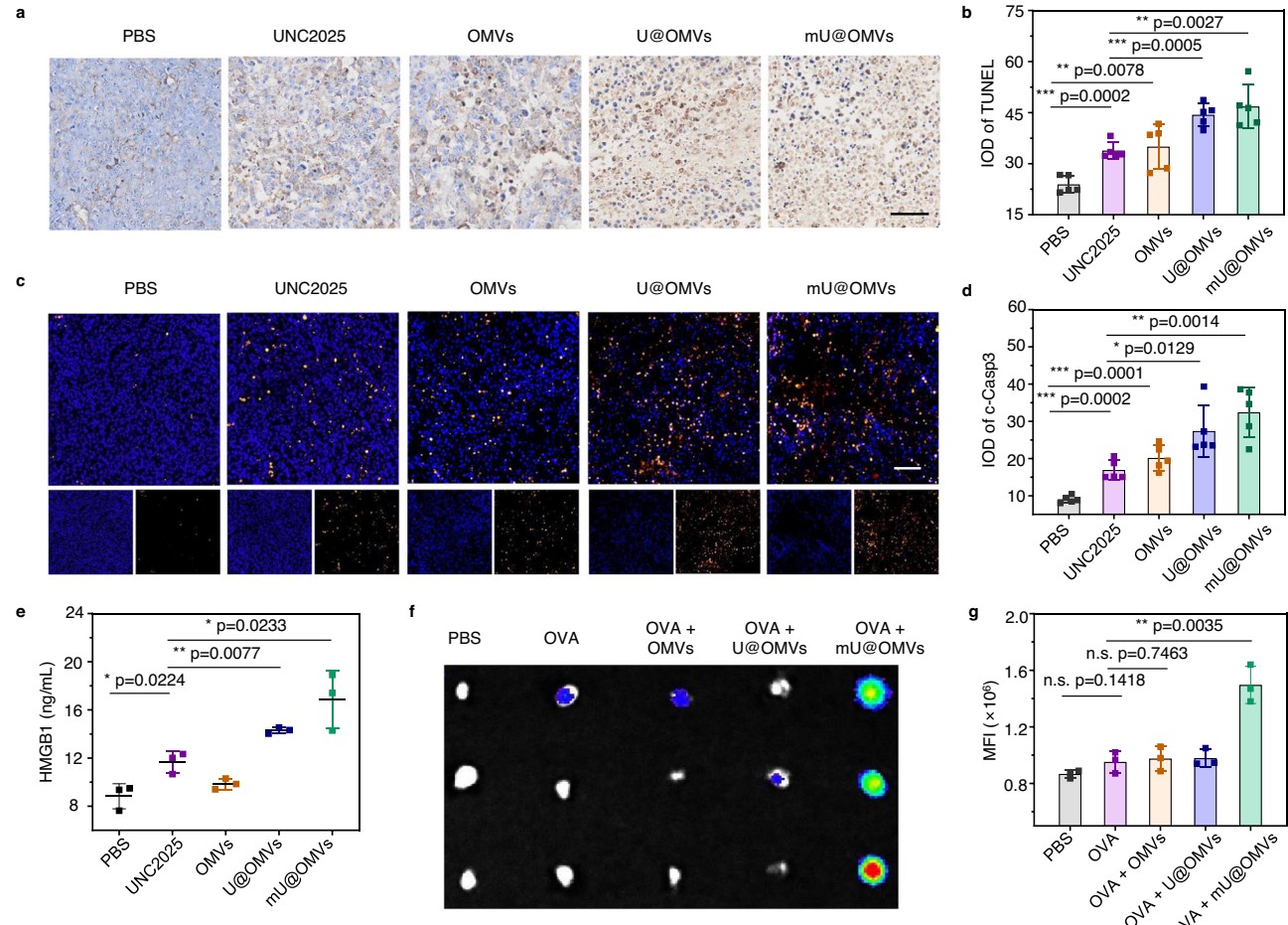

**Fig. 4 | Efferocytosis suppression-induced accumulation of AC and antigen delivery to LNs by mU@OMVs in vivo. a, c** Representative fluorescence images of TUNEL and c-Casp3 staining of B16F10 tumor sections obtained at 48 h after the treatment. Scale bar in **a** 50 μm. Scale bar in **c** 100 μm. **b, d** Integrated optical density (IOD) of the tumor slices in **a** and **c. e** Serum HMGB1 levels of mice after different treatments determined by ELISA. **f** Ex vivo imaging of the DLNs collected 24 h after administration. Briefly, Cy5-labeled OVA were incubated with OMVs, U@OMVs, or mU@OMVs for 3 h at 37 °C and then intratumorally administrated to B16F10 melanoma-bearing mice, and the DLNs were collected 24 h after injection. Then, OVA signals in the isolated DLNs were analyzed by the in vivo imaging system (IVIS). **g** Quantitative statistics of mean fluorescence intensity (MFI) of Cy5 channel within the region of interest (ROI) of isolated DLNs. Data in **b, d** are presented as mean ± s.d. ($n = 5$). Data in **e, g** are presented as mean ± s.d. ($n = 3$ biologically independent samples). Statistically significant differences between groups were identified by unpaired two-tailed Student's $t$-test. ****$P < 0.0001$, ***$P < 0.001$, **$P < 0.01$, *$P < 0.05$. Source data are provided as a Source Data file.

release of tumoral contents including TAAs. We also measured the HMGB1 concentration in tumor tissues, and the levels of HMGB1 in mU@OMVs treated mice were significantly enhanced than those in other groups, consistent well with the plasma results (Supplementary Fig. 18).

The LNs constituted of massive resident and migratory immune cells are pivotal for antitumor immunity[40]. It has been proved that the delivery efficiency of the in situ antigens to LNs is closely related to the strength of TAAs-initiated immune responses[41]. Migratory DCs are believed to be deficient in delivering antigens to LNs because DCs in tumors are dysfunctional and rare, so only a very small fraction of the antigens can be taken up and delivered[42]. However, OMVs can effectively target to LNs due to their optimal size and the improved internalization by immune cells in LNs[38,43]. Thus, we then tested whether the antigens captured by mU@OMVs could be delivered to LNs with the help of OMVs. Cy5-labeled OVA were mixed with OMVs, U@OMVs, or mU@OMVs firstly, and then these formations or free OVA were intratumorally injected into B16F10-bearing C57BL/6 mice, and the draining lymph nodes (DLNs) were collected 24 h after injection. We found that free OVA alone could hardly target DLNs, and neither OMVs (OVA + OMVs group) nor U@OMVs (OVA + U@OMVs group) could improve the targeting ability of free OVA, because OVA itself was difficult to

cross through the tumor stroma and could hardly interact with OMVs or U@OMVs. Nevertheless, a notable accumulation of OVA in DLNs was observed in the OVA+mU@OMVs group, and its fluorescence intensity was almost 1.5-fold that of the other groups (Fig. 4f, g). These results confirmed that mU@OMVs could effectively transfer the in situ captured TAAs to DLNs, indicating their great potential in initiating an antitumor immune response.

## Systemic antitumor immune responses stimulated by mU@OMVs

To explore the activation of immune responses in vivo, B16F10 tumor-bearing C57BL/6 mice were peritumorally injected with PBS, UNC2025, OMVs, U@OMVs, or mU@OMVs (Fig. 5a). Then, the DLNs, spleens, and tumors of mice were extracted at 48 h post-administration. In comparison with the PBS control, the UNC2025 treatment showed a modest effect on inducing the DCs maturation, probably due to the limited secondary necrosis of tumor cells inducing the inefficient release of TAAs and the difficult migration of local antigens to LNs (Fig. 5b–d and Supplementary Figs. 19–21). OMVs or U@OMVs treatment enhanced the proportions of matured DCs to a certain degree, relating to the adjuvanticity of OMVs. Interestingly, mice treated with mU@OMVs showed a much stronger ability in promoting DCs

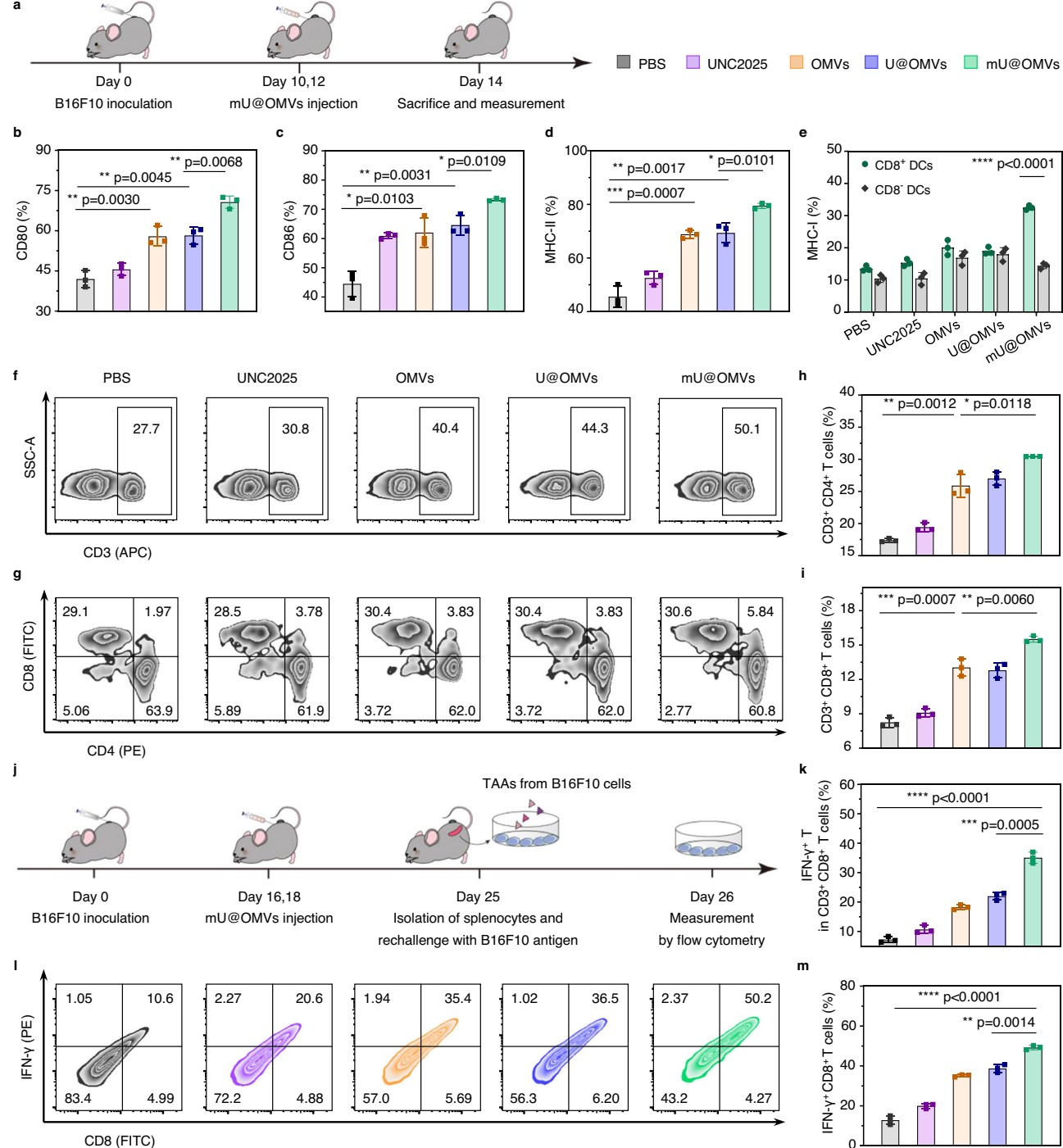

**Fig. 5 | Enhanced systemic immune responses by mU@OMVs in B16F10 tumor-bearing mice. a** Schematic illustration of experimental design. **b**–**d** Expression levels of CD80, CD86, and MHC-II gated on CD11c⁺ cells in DLNs by flow cytometry. **e** Percentages of MHC-I⁺ cells in CD11c⁺ CD8⁺ DCs and CD11c⁺ CD8⁻ DCs in DLNs. **f** Representative flow cytometry plots showing the CD3⁺ T cells in the spleens. **g**–**i** The percentages of CD4⁺ or CD8⁺ cells in CD3⁺ T lymphocytes detected by flow cytometry in the spleens. **j** Schematic illustration of the experiment for determination of antigen-specific immune responses. **k** IFN-γ secretion by splenocytes

(gated on CD3⁺ CD8⁺ T cells) after re-stimulation with B16F10 antigen for 24 h determined by flow cytometry. **l, m** Flow cytometry and quantification of expression levels of IFN-γ⁺ CD8⁺ T lymphocytes (gated on CD3⁺ cells) in tumors. Data in **b**–**e, h, i, k, m** are presented as mean ± s.d. (*n* = 3 biologically independent samples). Statistically significant differences between groups were identified by unpaired two-tailed Student's *t*-test. ****P < 0.0001, ***P < 0.001, **P < 0.01, *P < 0.05, n.s., not significant. Source data are provided as a Source Data file.

maturation. As a result, the CD80⁺, CD86⁺, MHC-II⁺, or MHC-I⁺ DCs in the DLNs significantly increased, emphasizing the ability of mU@OMVs to not only catch TAAs in tumors but also deliver them to LNs. In addition, resident CD8⁺ DCs are more efficient in cross-presenting tumor antigens for cytotoxic T lymphocyte responses[44,45].

Therefore, we further investigated the antigen cross-presenting ability of CD8⁺ DCs after different treatments. As shown in Fig. 5e and Supplementary Fig. 22, the MHC-I expression in CD8⁺ DCs was significantly enhanced than that in CD8⁻ DCs after mU@OMVs immunization, demonstrating that mU@OMVs could facilitate the cross-presentation

of tumor antigen by MHC class I molecules in resident CD8⁺ DCs, contributing to subsequent T-cell stimulation. As expected, mU@OMVs induced the highest proportion of CD3⁺ T cells, CD3⁺ CD4⁺ helper T cells, and CD3⁺ CD8⁺ T cells in the spleens (Fig. 5f, g and Supplementary Fig. 23). Particularly, the percentages of CD3⁺ CD4⁺ T cells and CD3⁺ CD8⁺ T cells in mU@OMVs group were individually increased by 13.2% and 7.3% in comparison with the PBS control, confirming the efficient elicitation of T cell responses in vivo (Fig. 5h, i). Next, we evaluated the antigen-specific immune response. The splenocytes from the immunized mice were collected and restimulated ex vivo with B16F10 antigen for 24 h. Next, the expression of IFN-γ in CD3⁺ CD8⁺ T cells was measured by flow cytometry (Fig. 5j, k and Supplementary Fig. 24). The proportions of IFN-γ⁺ T cells in the mU@OMVs groups were individually increased by 27.8%, 24.2%, 16.9%, and 13.1% compared with those in PBS, UNC2025, OMVs, and U@OMVs groups. These data suggested that mU@OMVs was an efficient in situ vaccine and could elicit a strong antigen-specific immune response in vivo. The serum levels of proinflammatory cytokines, including IL-6, TNF-α, IL-12p40, and interferon γ (IFN-γ) were also remarkably increased in this group, further indicating the systemic immune responses induced by mU@OMVs (Supplementary Fig. 25). Inspired by this, we then detected the antitumor immune responses in tumors. Impressively, the percentages of IFN-γ-secreted CD8⁺ T cells significantly enhanced in mU@OMVs treated mice, and the levels were individually increased by 36.4%, 29.2%, 14.1%, and 10.5% than that of PBS, UNC2025, OMVs, and U@OMVs groups, predicting the effective tumor-infiltrating cytotoxic T lymphocytes (CTL)-mediated immune responses that favorable for tumor elimination (Fig. 5l, m and Supplementary Fig. 26). Together, antigen-captured mU@OMVs could facilitate the DCs maturation and cross-presentation of tumor antigens by MHC class I molecules in resident CD8+ DCs, leading to the subsequent CD8⁺ T cell stimulation. A potent systemic and antigen-specific immune response was initiated in vivo under the smart synergism of UNC2025, Mal, and OMVs, suggesting its predictable efficacy in antitumor immunotherapy.

### Efficient antitumor effects of mU@OMVs

Encouraged by the above results, we next evaluated the antitumor efficacy of mU@OMVs in vivo. Firstly, a bilateral tumor model was established by successively inoculating B16F10 cells at the right flank (as the primary tumors) and left flank (as the abscopal tumors) of C57BL/6 mice as indicated in Fig. 6a. When the right tumors reached about $100 \, \text{mm}^3$, the mice were randomly divided into five groups and peritumorally injected with PBS, UNC2025, OMVs, U@OMVs or mU@OMVs at the primary tumors. The individual tumor volumes of primary and abscopal tumors were recorded every 3 days (Fig. 6b). We observed that UNC2025 alone could not suppress the tumor growth on either primary or distant tumors (Fig. 6c–f). OMVs or U@OMVs treatment could slow the tumor development of primary tumors but had limited therapeutic effects on distant tumors. The tumor control rates of these two groups were 60.73%, and 68.81% in primary tumors and only 19.98% and 13.75% in distant tumors, illustrating that without antigen capture and transfer, the growth of distant tumors could not be interfered with due to insufficient systemic immune responses as determined above. However, mU@OMVs strongly suppressed the development of both primary and distant tumors, and the tumor inhibitory rate reached almost 90% in primary tumors and 60% in abscopal tumors. The enhanced abscopal antitumor effect of mU@OMVs was associated with substantially increased tumor-infiltrating CD8⁺ T cells (almost 2 folds than other groups) (Fig. 6g and Supplementary Fig. 27). Moreover, the frequency of immunosuppressive regulatory T cells (Treg) significantly decreased to 5.2% in the abscopal tumors, which were respectively 6.8, 6.2, 3.3, and 2.7-fold lower than those of PBS, UNC2025, OMVs, and U@OMVs groups (Fig. 6h). Therefore, a significant increase of CD8⁺ T/Treg ratio (P-

values < 0.0001) was observed in the abscopal tumors, which were individually 14.7, 16.6, 5.2, and 4.7-fold higher than those of PBS, UNC2025, OMVs, and U@OMVs groups (Fig. 6i). Meanwhile, mU@OMVs group exhibited more extensive apoptotic cells on the bilateral tumor tissues as determined by the TUNEL staining, further proving its strong curative effects (Fig. 6j). As a result, the mice in other groups died within 30 days but nearly 80% of mice survived with negligible decreases of body weight in mU@OMVs group (Fig. 6k and Supplementary Fig. 28). Moreover, the analysis of hepatic function biomarkers including alanine transaminase (ALT), aspartate transaminase (AST), and alkaline phosphatase (ALP), renal function biomarkers including creatinine (Crea) and urea nitrogen (BUN), cardiac function biomarkers including lactate dehydrogenase (LDH) were all normal in mU@OMVs-treated mice (Supplementary Fig. 29a–f). Also, the hematoxylin and eosin (H&E) staining of the heart, liver, spleen, lungs, and kidneys showed no obvious damage, demonstrating the satisfactory safety profile of the mU@OMVs-based antitumor therapy (Supplementary Fig. 30).

We also assessed the therapeutic efficacy of mU@OMVs in a CT26 colon cancer model of mice as shown in Supplementary Fig. 31a. Consistently, the tumor volumes and tumor growth rates decreased in the order of PBS, UNC2025, OMVs, U@OMVs, and mU@OMVs (Supplementary Fig. 31b, c). mU@OMVs group achieved the most potent tumor inhibition with a tumor inhibitory rate of 82.05%, which was almost 9-fold that of UNC2025, and 2-fold that of OMVs or U@OMVs (Supplementary Fig. 31d). Also, mU@OMVs effectively enhanced the survival rate to 90% during 40 days, while mice in other groups gradually died within 30 days (Supplementary Fig. 31f). All these results verified the compelling therapeutic efficacy of mU@OMVs against tumors.

### Inhibition of tumor metastasis and tumor recurrence by mU@OMVs

Considering the robust immune responses and therapeutic effects achieved by mU@OMVs, we further evaluated its potential in treating tumor metastasis and recurrence. B16F10 tumor-bearing mice were randomly divided into five groups and treated with different formations peritumorally. 24 h after the last administration, $3 \times 10^5$ B16F10 cells were intravenously injected into the mice to mimic the escape of tumor cells from a primary tumor (Fig. 7a). After another 15 days, the lung tissues in different groups were collected for metastasis analysis by H&E staining. As shown in Fig. 7b, c, obvious lung metastatic foci were found in the PBS, UNC2025, OMVs as well as U@OMVs groups, and many pulmonary metastasis nodules were recorded in these groups. As expected, mU@OMVs significantly reduced lung metastasis, and the number of metastasis nodules decreased by ~60% compared with other groups, indicating its great potential in preventing metastatic tumors.

We further established a tumor rechallenge model to evaluate the long-term immune memory effect of mU@OMVs. BALB/c mice bearing CT26 tumors were treated with PBS or mU@OMVs by peritumoral administration, and all tumors were removed by surgery on the 15th day. Then the mice were rechallenged with the CT26 tumor cells 4 days post-treatment (Fig. 7d). As shown in Fig. 7e and Supplementary Fig. 32, the mice treated with PBS could not inhibit the development of rechallenged tumors. The tumor volumes were gradually close to $1000 \, \text{mm}^3$ within 17 days, and the percentage of tumor recurrence was 80%. On the contrary, the mice treated with mU@OMVs showed complete regression of the reinjected CT26 tumors. The quantitative analysis results showed that the level of CD3⁺ CD8⁺ cytotoxic T cells in mU@OMVs group was 3.11-fold higher than that of the PBS group (Fig. 7f–h). In addition, the frequency of effector memory T cell ($T_{EM}$) (CD3⁺ CD8⁺ CD44⁺ CD62L⁻) in mU@OMVs treated mice increased by 29.93% compared with that of PBS control, demonstrating that

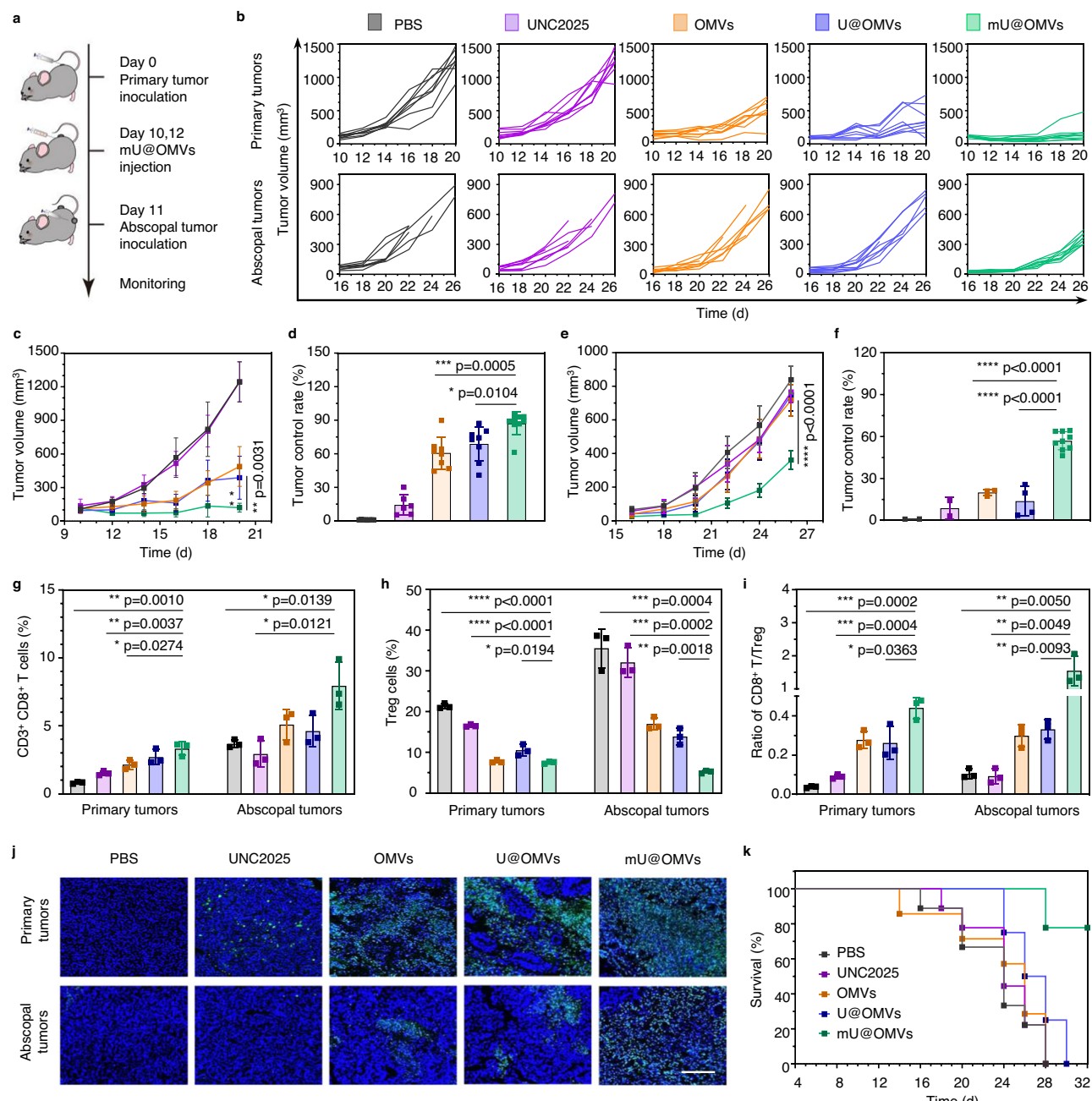

**Fig. 6 | The antitumor effects of mU@OMVs in inhibiting both primary and abscopal B16F10 tumors. a** Schematic illustration of mU@OMVs-mediated anti-tumor experiment in a bilateral tumor model. **b** Individual tumor growth kinetics of the primary and abscopal tumors are recorded every three days. **c, e** Tumor volumes of the primary tumors and abscopal tumors. **d, f** Tumor control rate of the primary tumors and abscopal tumors. **g** Percentages of CD3+ CD8+ T cell in primary and abscopal tumors. **h** Percentages of Treg cells in primary and abscopal tumors. **i** Ratios of CD8+ T cells to Treg cells in primary and distant tumors. **j** TUNEL staining of primary and abscopal tumors at the end of the experiment. Scale bar: 200 μm. **k** Survival curve of each group. Data in **b, c, e** are presented as mean ± s.d. ($n = 9$ biologically independent mice). Data in **g–i** are presented as mean ± s.d. ($n = 3$ biologically independent mice). Statistically significant differences between groups were identified by unpaired two-tailed Student's $t$-test. ****$P < 0.0001$, ***$P < 0.001$, **$P < 0.01$, *$P < 0.05$. Source data are provided as a Source Data file.

mU@OMVs could induce potent long-term immune memory effects to prevent tumor recurrence.

## Discussion

In this work, we constructed a versatile OMVs-based nanosystem mU@OMVs that could not only induce the second necrosis of tumor cells by inhibiting their physiological efferocytosis but also form a powerful in situ cancer vaccine by capturing the subsequently released whole-cell tumor-associated antigens (TAAs). Furthermore, the in situ cancer vaccine could transfer from tumors to the lymph nodes by taking advantage of the appropriate size of OMVs. In the immune cells-enriched lymph nodes, the strong recognition between PAMPs on OMVs and PRRs on DCs facilitated the internalization of TAAs by DCs. Then, the smart synergism of the abundant TAAs with adjuvant OMVs boosted the maturation of DCs and activated the antigen presentation by MHC-I molecules on resident CD8+ DCs, finally evoking robust tumor-specific immunity that suppressed the established bilateral tumors as well as prevented the tumor metastasis and recurrence.

This versatile nanosystem with potent therapeutic effects may be universally applied. Firstly, both apoptosis and spontaneous

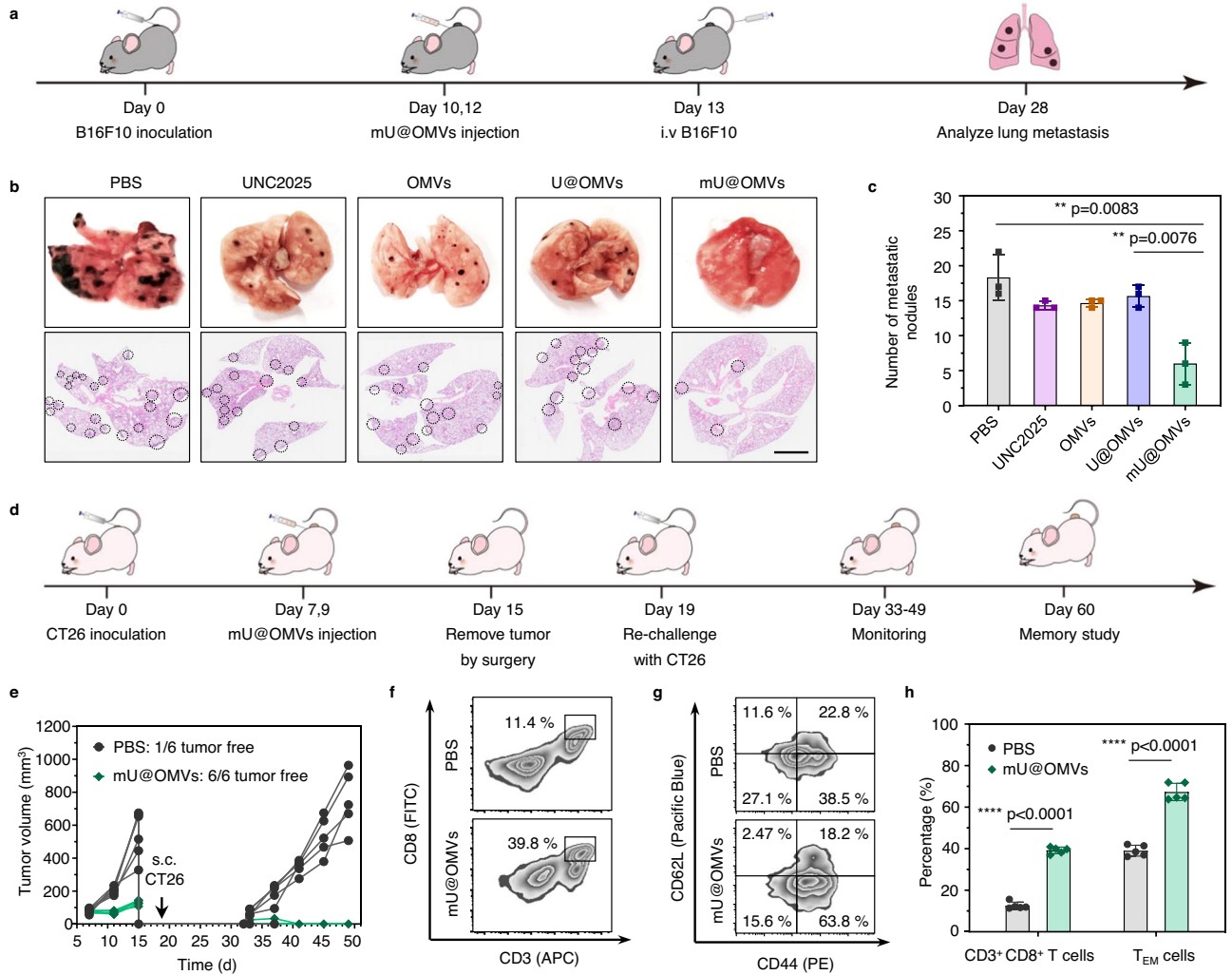

**Fig. 7 | Prevention of tumor metastasis and long-term memory effects in tumor recurrence. a** Schematic illustration of mU@OMVs-triggered cancer immunotherapy in a B16F10 metastasis model. **b** Representative H&E-stained lung slices. The metastatic nodules were outlined with black circles. Scale bar: 200 μm. **c** Numbers of lung metastatic nodules in each group. **d** Schematic illustration of treatment schedule for mU@OMVs-mediated prevention of tumor recurrence in a CT26 tumor model. **e** Tumor growth curves. **f** CD3⁺ CD8⁺ T lymphocytes in spleens detected by flow cytometry on day 60. **g** Flow cytometry data of CD3⁺ CD8⁺ CD44⁺ CD62L⁻ (TEM) T lymphocytes in spleens detected by flow cytometry on day 60. **h** quantification of expression levels of CD3⁺ CD8⁺ T lymphocytes and CD3⁺ CD8⁺ CD44⁺ CD62L⁻ T lymphocytes ($T_{EM}$) in spleens. Data in **c** are presented as mean ± s.d. ($n = 3$ biologically independent mice). Data in **h** are presented as mean ± s.d. ($n = 5$ biologically independent mice). Statistically significant differences between groups were identified by unpaired two-tailed Student's $t$-test. ****$P < 0.0001$, ***$P < 0.001$, **$P < 0.01$, *$P < 0.05$. Source data are provided as a Source Data file.

efferocytosis are widespread in tumors, and the application of this nanosystem has no restrictions on individual differences, tumor types, and heterogeneity, even promising for the development of personalized tumor vaccines. In addition, this in situ antigen capture strategy holds great potential in combination with various clinical treatments, such as chemotherapy, radiotherapy, and phototherapy, since all these therapeutics can promote in situ antigen generation. As a novel example, Li et al. used the maleimide group-modified OMVs to capture large amounts of in situ antigens after photothermal therapy[30]. Furthermore, considering the massive efferocytosis of apoptotic cells by TAMs that followed the cytotoxic therapeutics, our efferocytosis blockade-based vaccine strategy can synergistically avoid the engulfment of these apoptotic cells, further enhancing the antitumor immune response. Moreover, the nano-sized bacterial OMVs are able to be large-scale prepared and easily be genetically engineered or chemically modified. A universal OMVs-based nanosystem platform would be provided for different therapeutic purposes.

We can believe that OMVs-based diagnosis and treatment systems are of great potential in clinical practice. Particularly, an OMVs-based vaccine against Neisseria meningitidis has been recently approved by the European Commission. In the future, the safety concerns should be further addressed for the wider application. Therefore, new non-pathogenic bacteria should be developed to produce OMVs, and new strategies including surface modification and gene encoding, that can reduce the side effects of OMVs, should be exploited. As for our mU@OMVs system, the therapeutic effects would be further augmented if mU@OMVs cooperated with the antibodies of immune checkpoint programmed death 1 ligand 1 (PD-L1) or inhibitors against indoleamine-2,3-dioxygenase activity 1 (IDO1) since efferocytosis blockade and OMVs-mediated immune response will induce the expression of interferon-γ (IFN-γ), which is known to upregulate the immunosuppressive factors like PD-L1 and IDO1.

## Methods
### Materials and reagents
DMEM, RPMI-1640 cell culture medium, fetal bovine serum (FBS), and 0.25% trypsin-ethylenediaminetetraacetic acid (EDTA) were purchased from Gibco (USA). Penicillin−streptomycin was purchased from

Macgene (China). UNC2025·2HCl was purchased from Topscience (China). Mal-PEG4-NHS was purchased from QIYUE BIOLOGY (China). Ovalbumin (OVA), PE-conjugated OVA, Cy5-conjugated OVA, FITC-conjugated anti-*E. coli* LPS antibody (catalog no. bs-8000R-FITC), anti-phospho-MerTK antibody (catalog no. bs-18791R), anti-phospho-Axl antibody (catalog no. bs-5181R), and anti-Axl antibody (catalog no. bs-5180R) were purchased from Bioss (China). Anti-MerTK (catalog no. ab270448), anti-β-Actin (catalog no. ab8227), and Goat-Anti-Rabbit IgG (HRP) (catalog no. ab6721) antibodies were purchased from Abcam (UK). FITC-conjugated anti-mouse CD11c (catalog no. AB_2883792) antibody was purchased from Proteintech (China). APC anti-mouse CD11b (catalog no. 101211), FITC-conjugated anti-mouse CD80 (catalog no. 104705), PE-conjugated anti-mouse CD86 (catalog no. 159203), APC-conjugated anti-mouse MHC-II (catalog no. 107613), PE-conjugated anti-mouse MHC-I (catalog no. 114608), PE-conjugated anti-mouse CD69 (catalog no. 104507), APC-conjugated anti-mouse CD3 (catalog no. 100312), FITC-conjugated anti-mouse CD8 (catalog no. 100705), PE/FITC-conjugated anti-mouse CD4 (catalog no. 100405), PE-conjugated anti-mouse CD25 (catalog no. 101903), Pacific Blue-conjugated anti-mouse Foxp3 antibody (catalog no. 126409), PE-conjugated anti-mouse IFN-γ (catalog no. 505807), PE-conjugated anti-mouse CD44 (catalog no. 103007), Pacific Blue-conjugated anti-mouse CD62L antibody (catalog no. 104424), Mouse IL-6, IL-12p40, TNF-α, IFN-γ ELISA Kit were purchased from BioLegend (USA). PE-conjugated anti-mouse CD206 antibody (catalog no. 12-2069-42) and flow cytometry staining buffer were purchased from eBioscience (USA). Carboxyfluorescein succinimidyl ester (CFSE) was purchased from Beyotime (China). Mouse high mobility group protein B1 (HMGB1) ELISA Kit and the Annexin V-FITC/DAPI Apoptosis Detection Kit were purchased from Elabscience (China). Micro BCA Protein Assay Kit and Prestained Protein Ladder were purchased from Thermo Scientific (USA). Coomassie brilliant blue, Cell Counting Kit-8 (CCK-8), Silver stain kit, and SDS-PAGE loading buffer were purchased from Solarbio (China). Hoechst 33342 was purchased from Life Technologies (USA). Recombinant murine IL-4 and granulocyte−macrophage colony-stimulating factor (GM-CSF) were purchased from PeproTech (USA).

### Cell lines and animals
CT26 (mouse colon cancer cell line) and B16F10 (murine melanoma cancer cell line) were kindly obtained from the Institute of Process Engineering (China). RAW264.7 (murine macrophage cell line) was purchased from Peking Union Medical College Hospital (catalog number: 1101MOU-PUMC000146). All cell lines were maintained in a 37 °C humidified chamber with 5% $CO_2$. CT26, RAW264.7, and B16F10 cells were cultured in a DMEM medium containing 10% FBS and 1% penicillin−streptomycin.

Female BALB/c mice and C57BL/6 mice were purchased from Beijing Vital River Laboratory Animal Technology Co. Ltd (Beijing, China). Mice used in these studies were 6 weeks old at the start of the experiment. Mice were housed in a room with a temperature of 20−22 °C and a humidity of 30−70%. Feed and water were available. Artificial light was provided in a 12-h light/12-h dark cycle. All mice received human care according to the animal care regulations of the Beijing Animal Ethics Association and the Ethics Committee of the Beijing Institute of Technology (approval ID: 2019-0010-M-2020019).

### Bacteria strains
*Escherichia coli* MG1655 was purchased from China Center for Type Culture Collection (CCTCC).

### Preparation of bacterial outer membrane vesicles (OMVs)
*Escherichia coli* MG1655 in LB medium was incubated in a shaking incubator at 220 rpm at 37 °C overnight. Then, bacterial cells were removed by centrifuging at $10,000 \times g$ for 10 min at 4 °C (Sorvall ST16R, Thermo Scientific). The obtained supernatant was filtered by 0.45 μm filters (Vacuum Filter System, Corning) and concentrated by centrifugal filters with a molecular weight cutoff (MWCO) of 100 kDa (Millipore). The concentrated supernatant was filtered again through 0.22 μm pore size filters (Millipore) to remove any remaining debris or bacteria. Then, the concentrated medium was centrifuged at $150,000 \times g$ for 3 h at 4 °C (XPN 100, Beckman). The supernatant was removed and the pellet was resuspended in PBS and stored at -80 °C. The protein concentration of OMVs was determined by Micro BCA Protein Assay Kit.

### Preparation of U@OMVs and mU@OMVs
To obtain U@OMVs, UNC2025·2HCl molecules were encapsulated into OMVs through electroporation. In brief, 10 μg UNC2025·2HCl and 10 μg OMVs were gently suspended in the electroporation cuvette on ice. Electroporation was operated under conditions of 100 V, 200 Ω, and 100 μF by the BTX ECM 630 and GenePulser electroporators (Bio-Rad). After that, residual UNC2025·2HCl was removed by ultra-centrifugation at $5000 \times g$ for 10 min and washed three times. The loading efficiency and encapsulation efficiency of U@OMVs were determined by UV-vis spectrum. To obtain mU@OMVs, 200 μL U@OMVs (containing 10 μg protein) were mixed with 10 μg Mal-PEG4-NHS solution and incubated at room temperature for 3 h. Excess Mal-PEG4-NHS was removed by ultracentrifugation at $150,000 \times g$ for 3 h at 4 °C. The pellet containing mU@OMVs was resuspended in PBS and the content of UNC2025·2HCl in the supernatant was determined by UV-vis spectrum to ensure the variety of loading efficiency and encapsulation efficiency.

### Characterization of OMVs, U@OMVs, and mU@OMVs
The sample morphology of OMVs, U@OMVs, or mU@OMVs was observed by the transmission electron microscope (TEM) (120 kV, Tecnai Spirit). The hydrodynamic sizes and zeta potentials were measured by dynamic light scattering (DLS) (Malvern Instruments). Nanoparticle tracking analysis (NTA) was also used to measure the size distribution of OMVs, U@OMVs, and mU@OMVs (Particle Metrix, zetaview). The optical properties were characterized by the UV-vis absorbance spectra (Multiskan Sky, Thermo Scientific).

### In vitro antigen capture test
A 500 μL OMVs, U@OMVs, or mU@OMVs solution (containing 50 μg protein) was mixed with 50 μg of OVA in a phosphate buffer (pH 6.8) and incubated for 4 h under gentle shaking at room temperature. The OVA-adsorbed vesicles were separated from free OVA by ultracentrifugation at $150,000 \times g$ for 3 h at 4 °C. The pellet was resuspended with 100 μL PBS and their protein concentrations was determined by BCA assay. Then, samples were mixed with loading buffer, and heated at 99 °C for 15 min to denature the proteins. After SDS-PAGE electrophoresis, proteins were stained with coomassie brilliant blue for 2 h and then imaged to further analyze the proteins in these vesicles.

### In vitro cytotoxicity
Cytotoxicities of free UNC2025 and mU@OMVs were determined by CCK-8 assays. Briefly, CT26, RAW264.7, or B16F10 cells were seeded in 96-well microplates at a density of $1.5 \times 10^4$ cells/well and incubated overnight at 37 °C. Then, free UNC2025 or mU@OMVs with different concentrations were added to the wells. After 24 h treatment, cells were incubated with CCK-8 solution for another 1 h. Finally, the absorbance at 450 nm for each well was measured by UV-spectrophotometry (Multiskan Sky, Thermo Scientific).

### Cellular uptake of OMVs, U@OMVs, and mU@OMVs
RAW264.7 cells ($1 \times 10^5$ cells/well) were seeded in a 24-well plate and treated with 100 ng/mL IL-4 to polarize M0 to M2 macrophages for 48 h. Then, the marker of M2 macrophages (CD206) was labeled with

PE-conjugated anti-mouse CD206 antibody. Finally, the labeled cells were detected by a flow cytometer (Bioscience FACSAria, BD).

To determine the cellular uptake, OMVs were prelabeled with FITC-conjugated LPS antibody and incubated at 37 °C for 1 h. The free antibody was removed by centrifugal filters with MWCO of 300 kDa (Pall). U@OMVs and mU@OMVs were prepared as previously mentioned using prelabeled OMVs. Then, OMVs, U@OMVs, or mU@OMVs (30 μg/mL) were added to M2 macrophages and incubated for 4, 3, 2, 1, and 0.5 h, respectively. Afterward, parts of the cells were stained with Hoechst 33342 (100 μM in PBS) for 15 min at 25 °C. The final M2 macrophages were detected by flow cytometer (Bioscience FACSAria, BD) or confocal microscopy (Eclipse-Ti2, Nikon), respectively.

### Western Blot analysis of the inhibition of MerTK phosphorylation

RAW264.7 cells ($2 \times 10^5$ cells/well) were seeded in a 12-well plate and treated with 100 ng/mL IL-4 to polarize M0 to M2 macrophages for 48 h. Then, free UNC2025 or mU@OMVs with indicated dosages were added to M2 macrophages and incubated for 4, 2, 1, and 0.5 h, respectively.

Protein samples extracted from the above cells were separated with SDS polyacrylamide gel electrophoresis and blotted onto nitrocellulose (NC) membrane (Millipore). The membrane was blocked with 5% BSA for 3 h and incubated with anti-MerTK antibody (1:1000), anti-phospho-MerTK (1:2000), or anti-β-Actin antibody (1:10,000) overnight at 4 °C. After washing with phosphate-buffered saline with Tween20 (PBST) for three times, the membrane was further incubated with a diluted secondary antibody Goat-Anti-Rabbit IgG (HRP) (1:10,000) for 2 h. Then the membrane was washed by PBST three times and visualized with ECL reagent (Thermo Scientific) by Chemi-luminescence system (ChemiDoc XRS+ System, Bio-Rad).

### In vitro efferocytosis assay

Apoptosis of B16F10 tumor cells (AC) was induced by 5 μM doxorubicin at 37 °C for 6 h. Exposure of phosphatidylserine on the cell surface was assessed using the Annexin V-FITC/DAPI Apoptosis Detection Kit. Apoptotic B16F10 tumor cells were then labeled with 5 μM CFSE at 37 °C for 20 min. M2 macrophages were pre-incubated with PBS, OMVs (3.6 μg/mL), UNC2025 (1 μg/mL), U@OMVs (containing 1 μg/mL UNC2025), mU@OMVs (containing 1 μg/mL UNC2025) 2 h prior to adding CFSE-labeled apoptotic B16F10 cells. After 6, 12, and 18 h incubation, macrophages in mixed cells were labeled with APC-conjugated anti-CD11b antibody. Then, cells were collected and analyzed by flow cytometer (Bioscience FACSAria, BD) or confocal microscopy (Eclipse-Ti2, Nikon). The supernatants from co-culture were collected to examine the HMGB1 level by ELISA.

The protein concentration of the above supernatants of M-AC + mU@OMVs groups after the 18 h culture was determined by the Micro BCA Protein Assay Kit. Then, a 1 mL mU@OMVs solution (containing 200 μg protein) was mixed with 800 μg of supernatants in a phosphate buffer (pH 6.8) and incubated under gentle shaking at room temperature. The proteins-adsorbed mU@OMVs were separated from free proteins by ultracentrifugation at $5000 \times g$ for 10 min and washed three times. The obtained proteins-adsorbed mU@OMVs were collected, and a proteomics experiment was performed to determine the adsorbed peptide/protein on the mU@OMVs. The relative abundance of proteins captured by mU@OMV was calculated by dividing the values of each peptide intensity by the value of the peptide with the highest intensity (peptide complement C3), meanwhile, the value of the relative abundance of peptide C3 was defined as 10.

### Dendritic cellular uptake of OVA-adsorbed mU@OMVs

Murine bone marrow-derived dendritic cells (BMDCs) from marrow cavities of femurs and tibias of C57BL/6 mice were cultivated in plates with a medium containing 20 ng/mL GM-CSF and 20 ng/mL IL-4 for 7 days. Then, BMDCs were seeded in a 24-well plate at a density of $1 \times 10^6$ per well.

mU@OMVs solution was mixed with PE-labeled OVA in a phosphate buffer (pH 6.8) and incubated for 4 h under gentle shaking at room temperature. For uptake studies, BMDCs were incubated with free PE-labeled OVA or mU@OMVs pre-mixed with PE-labeled OVA (concentration of OVA = 10 μg/mL) for 12 h at 37 °C. Then, BMDCs were labeled with FITC-conjugated anti-CD11c antibodies. The Uptake was measured by the mean fluorescence intensity (MFI) of the FITC signal in $CD11c^+$ cells via flow cytometry (Bioscience FAC-SAria, BD). For uptake imaging, DCs were also stained with Hoechst 33342 and then imaged with a confocal microscope (Eclipse-Ti2, Nikon).

### Analysis of BMDCs maturation and T cell activation

Murine bone marrow-derived dendritic cells (BMDCs) from marrow cavities of femurs and tibias of C57BL/6 mice were cultivated in plates with a medium containing 20 ng/mL GM-CSF and 20 ng/mL IL-4 for 7 days. Then, BMDCs were seeded in a 24-well plate at a density of $5 \times 10^5$ per well.

The supernatants of necrotic B16F10 cells induced by doxorubicin were incubated with OMVs, U@OMVs, or mU@OMVs for 4 h under gentle shaking at room temperature, and respectively, added into BMDCs for 24 h (final concertation of OMVs = 5 μg/mL). Then, BMDCs were collected and stained with fluorescent antibodies of CD11c, CD80, CD86, MHCI, and MHCII individually, and measured by flow cytometry (Bioscience FACSAria, BD). For the quantitative analysis of cytokines, the supernatants of BMDCs were collected and analyzed using IL-6, IL-12p40, and TNF-α ELISA kits.

Splenocytes were isolated from the spleens of C57BL/6 mice (aged 6−8 weeks) and $CD8^+$ T cells were isolated using a $CD8^+$ no-touch isolation kit (Miltenyi Biotec) according to the manufacturer's guidelines. Afterward, T cells were coincubated with the above-treated BMDCs for 24 h at a ratio of 1:4. Then, T cells were collected and labeled with APC-conjugated anti-mouse CD3, FITC-conjugated anti-mouse CD8 as well as PE-conjugated anti-mouse CD69, and analyzed by flow cytometer (Bioscience FACSAria, BD).

### In vivo apoptosis and immune activation assay

C57BL/6 mice were subcutaneously incubated with B16F10 cells ($1 \times 10^6$). When tumor volumes reached 100 mm³, mice were randomly divided into five groups and peritumorally injected with 100 μL of PBS, UNC2025, OMVs, U@OMVs, or mU@OMVs every 3 days for two times (OMVs dose: 10 μg per mouse; UNC2025 dose: 2.7 μg per mouse). After 48 h, mice were sacrificed, and the draining lymph nodes (DLNs), spleens, and tumor tissues were surgically collected for further investigation. Sera were collected for quantitative analysis of IL-6, IL-12p40, TNF-α, IFN-γ, and HMGB1. Tumors were fixed in 4% paraformaldehyde solution and sectioned into slices for the terminal deoxynucleotidyl transferase-mediated deoxyuridine triphosphate nick end labeling (TUNEL) and the cleaved-Caspase 3 (c-Casp3) staining and imaged by a slide scanner (VS200, Olympus).

For immune cell analysis, DLNs, spleens, and tumors were cut into small pieces and then homogenized in cold PBS to form the single-cell suspension. The obtained cells were divided into several parts to analyze different immune cell types, respectively. For DC maturation analysis, cells from DLNs were stained with CD11c, CD80, CD86, MHC-II, and MHC-I antibodies. For $CD8^+$ T cells analysis, cells from spleens were stained with CD3, CD4, and CD8 antibodies. For cytotoxic T lymphocytes (CTL) analysis, cells from tumors were stained with CD3, CD8, and IFN-γ antibodies. Then, stained cells were washed with PBS three times and measured by flow cytometer (Bioscience FACSAria, BD).

## Antigen-specific immune response

C57BL/6 mice were subcutaneously incubated with B16F10 cells ($1 \times 10^6$). When tumor volumes reached 100 mm³, mice were randomly divided into five groups and peritumorally injected with 100 μL of PBS, UNC2025, OMVs, U@OMVs, or mU@OMVs every 3 days for two times (OMVs dose: 10 μg per mouse; UNC2025 dose: 2.7 μg per mouse). On day 25, the splenocytes from the immunized mice were collected and restimulated ex vivo with B16F10 antigen (obtained by multigelation) for 24 h. Next, the expression of IFN-γ in CD3$^+$ CD8$^+$ T cells was measured by flow cytometry.

## Antigen delivery of mU@OMVs in vivo

5 μg Cy5-labeled OVA were mixed with 10 μg OMVs, U@OMVs, or mU@OMVs, and incubated for 4 h under gentle shaking at room temperature. C57BL/6 mice were subcutaneously incubated with B16F10 cells ($1 \times 10^6$). When tumor volumes reached 200 mm³, mice were randomly divided into five groups and intratumorally injected with 100 μL of PBS, free Cy5-labeled OVA, a mixture of OMVs and Cy5-labeled OVA, a mixture of U@OMVs and Cy5-labeled OVA, or a mixture of mU@OMVs and Cy5-labeled OVA, respectively (OVA dose: 5 μg per group, multipoint injection). After 24 h, the fluorescence of DLNs was monitored by an in vivo imaging system (IVIS Spectrum, PerkinElmer).

## In vivo antitumor activity

To establish a bilateral tumor model, $1 \times 10^6$/100 μL B16F10 tumor cells in PBS were subcutaneously transplanted into the right flank of C57BL/6 mice as the primary tumor. After 11 days, to form the abscopal tumor, $1 \times 10^6$ B16F10 was subcutaneously injected into the left flank. When the primary tumor volume reached about 100 mm³, mice were randomly divided into five groups, and peritumorally injected with 100 μL of PBS, UNC2025, OMVs, U@OMVs, or mU@OMVs every 3 days for two times at primary tumors (OMVs dose: 10 μg per mouse; UNC2025 dose: 2.7 μg per mouse). The length ($L$) and width ($W$) of the subcutaneous tumors and the body weights were measured every other day after the first administration. The tumor volumes were calculated by the formula of ($L \times W^2$)/2. When the tumor volume was larger than 1500 mm³, the mice were euthanized by cervical dislocation. At the end of therapy, some mice were sacrificed and the main organs (heart, liver, spleen, lung, and kidneys), as well as bilateral tumors, were harvested, fixed in 4% paraformaldehyde solution, and sectioned into slices for the hematoxylin and eosin (H&E) or TUNEL staining. For safety evaluation, the serum levels of alkaline phosphatase (ALP), alanine aminotransferase (ALT), aspartate transaminase (AST), blood urea nitrogen (BUN), lactate dehydrogenase (LDH), and creatinine (CREA) at the endpoint of the experiment were analyzed.

For CT26 tumor inhibition, BALB/c mice were subcutaneously incubated with CT26 tumor cells ($1 \times 10^6$). When tumors reached about 100 mm³, mice were randomly divided into five groups, and peritumorally injected with 100 μL of PBS, UNC2025, OMVs, U@OMVs, or mU@OMVs every 3 days for two times (OMVs dose: 10 μg per mouse; UNC2025 dose: 2.7 μg per mouse). The tumor volumes were calculated by the formula of ($L \times W^2$)/2. The body weight was measured every 2 days after the first administration.

## T-cell infiltration in the tumor microenvironment

To establish a bilateral tumor model, $1 \times 10^6$/100 μL B16F10 tumor cells in PBS were subcutaneously transplanted into the right flank of C57BL/6 mice as the primary tumor. After 11 days, to form the abscopal tumor, $1 \times 10^6$ B16F10 was subcutaneously injected into the left flank. When the primary tumor volume reached about 100 mm³, mice were randomly divided into five groups, and peritumorally injected with 100 μL of PBS, UNC2025, OMVs, U@OMVs, or mU@OMVs every 3 days for two times at primary tumors (OMVs dose: 10 μg per mouse; UNC2025 dose: 2.7 μg per mouse). The primary and abscopal tumors were isolated and analyzed on the 7th day after the final administration of PBS, UNC2025, OMVs, U@OMVs, or mU@OMVs.

## Evaluation of metastasis and recurrence

To construct the lung metastasis model, $1 \times 10^6$/100 μL B16F10 tumor cells were transplanted into the flank of the mice. When tumors reached about 100 mm³, mice were randomly divided into five groups, and peritumorally injected with different formulations like before. After 24 h, the mice were intravenously injected with B16F10 cells ($3 \times 10^5$). After another 15 days, the mice were killed, and their lungs were excised. Lung metastasis nodules were then manually counted, and lung tissue sections were subjected to H&E staining.

To construct the recurrence model, BALB/c mice were subcutaneously transplanted with $1 \times 10^6$ CT26 tumor cells. When tumors reached about 100 mm³, mice were randomly divided into five groups, and peritumorally injected with different formulations like before. On day 15, tumors in different groups were removed by surgery. On day 19, the mice were challenged by subcutaneous $1 \times 10^6$ CT26 tumor cells. Tumor growth was evaluated every four days. On day 60, splenocytes were isolated and the percentage of memory T cells (CD3$^+$ CD8$^+$ CD44$^+$ CD62L$^-$) was detected by flow cytometry (Bioscience FACSAria, BD).

## Statistical analysis

Data were reported as mean ± SD. An unpaired two-tailed Student $t$-test was used to analyze the statistically significant differences and data were considered statistically significant when the values of $P < 0.05$. *$P < 0.05$, **$P < 0.01$, ***$P < 0.001$, and ****$P < 0.0001$. n.s., not significant. Image Lab (version number, 3.0) was used to analyze the data of western blot. FlowJo (version number, 10.0.0.0) was used to analyze the data of flow cytometry. GraphPad Prism (version number, 8.0.2.263) was used for the statistical analysis. Living Image software (version number, 4.3.1.16427) was used to analyze the data of in vivo bioluminescence assay.

## Reporting summary

Further information on research design is available in the Nature Portfolio Reporting Summary linked to this article.

## Data availability

The authors declare that all the data supporting the findings of this study are available within the article and Supplementary Information. Source data are provided with this paper.

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

## Acknowledgements

This work was funded by the National Science Fund for Distinguished Young Scholars (22025401 to H.-Y.X.), the National Natural Science Foundation of China (22293034 to H.-Y.X., 22274011 to L.-L.H., 32101140 to W.N.) and the Beijing Institute of Technology Research Fund Program for Young Scholars (XSQD-202212002 to W.N.). The authors thank Biological & Medical Engineering Core Facilities (Beijing Institute of Technology) for providing advanced equipments.

## Author contributions

W.-R.Z. and H.-Y.X. conceptualized and designed the research. W.-R.Z., Y.W., C.L., J.H., and L.Z. performed the experiments. W.-R.Z., W.N., Y.L., and L.-L.H. collected and processed the data. All authors analyzed and interpreted the data. W.-R.Z. and H.-Y.X. wrote the manuscript. All authors discussed the results and commented on the manuscript.

## Competing interests

The authors declare no competing interests.
