## [Peer Review File · Nature Communications]

Bacterial outer membrane vesicle based versatile nanosystem boosts the efferocytosis blockade triggered tumor-specific immunityREVIEWER COMMENTS

Reviewer #1 (Remarks to the Author):

In this manuscript, the authors developed a versatile nanosystem for efficient tumor immunotherapy by synergizing the efferocytosis blockade-induced secondary necrosis with the bacterial outer membrane vesicles (OMVs)-based antigen transfer and immune augment. mU@OMVs effectively inhibited exocytosis, prevented MerTK phosphorylation in tumor-associated macrophages (TAMs), and captured the released tumor-associated antigens (TAAs). Moreover, OMVs are loaded with antigen and then drain to lymph nodes, thereby triggering a strong T-cell immune response and inhibiting tumor progression in multiple tumor models. I am optimistic for this manuscript and recommend that the study should be published after addressing the following major suggestions.

1. The main concern is the novelty of the research. The construction and design of vectors based on OMVs and the concept of "in situ antigen capture" are similar to the published articles (Li et al, Small, 2022).

Li et al (Small, 2022) presented an OMV-based multifunctional in situ vaccine with capacities of antigen capture and immune modulation to enhance the immune-mediated tumor clearance after PTT. Briefly, the surface of native OMVs was modified with maleimide groups (Mal) (OMV-Mal) to capture large amount of tumor antigens after PTT, while 1-methyl-tryptophan (1-MT, an inhibitor of indoleamine 2, 3-dioxygenase (IDO)) was loaded in OMV-Mal into OMV-Mal (1-MT@OMV-Mal) by electroporation to cover the immunosuppressive microenvironment mediated by Tregs. In situ injection of 1- MT@OMV-Mal simultaneously overcomes the immune inhibition of IDO on tumor-infiltrating effector T cells, leading to remarkable inhibition on both primary and distant tumors. Please compare the differences and advantages of the two studies in detail in the discussion and add it as reference.

2. The gating strategy for the flow cytometry analyses should be provided.

3. In addition to T cells, key immunosuppressive cells in tumor tissue should be analyzed. At the same time, the immune microenvironment in the distal tumor should be analyzed simultaneously.

4. When OMVs capture tumor antigen, it is often taken up by local APCs and transported to draining lymph nodes in a cell-dependent manner. In this study, this process was not characterized. In addition, does inhibition of macrophage efferocytosis affect their ability to take up OMVs and antigen?

5. In figure 1h, in addition to the OVA bands, there were some different protein bands between OMVs and group 3-5. Please explain.

6. In figure 3j, CD69 is a non-specific activation marker of T cells. In addition, the INF γ + CD8 T cells are also not antigen-specific T cells. Please add antigen specific experiments to prove the antigen capture of mU@OMVs. For example, the INF γ + CD8 T cells should be detected after antigen re-challenge using ELISPOT.

Reviewer #2 (Remarks to the Author):

In this manuscript, the authors construct a nanosystem composed of bacterial outer membrane vesicles loaded with UNC2025 and modified with maleimide on the surface. This nanosystem can not only inhibit efferocytosis but also boost the following antitumor immunity, which efficiently prevents the growth, metastasis, and recurrence of tumors in mice.

Here are some issues that need to be explained or revised about this project:

1. Why the nanoparticles were administrated peritumorally, but not intravenously or intratumorally?

2. As originated from the biomembrane, OMVs are rich in thiol, which makes it unreasonable to use DSPE-PEG-Mal for surface modification. How to ensure and characterize that DSPE-PEG-Mal is

inserted into the membrane through hydrophobic interaction, rather than the covalent bond between thiol and Mal? Besides, even if Mal can be modified on the surface of OMVs by the pattern described in this manuscript. I am afraid the OMVs would be cross-linked, therefore triggering the dramatic nanoparticle size increase observed in Figure 1.

3. Why did the article size increase so much after being encapsulated with UNC2025?
4. As a potentially pathogenic bacterium, is the E. coli used for the extraction attenuated? If not, more safety evaluations are needed, such as the investigation of endotoxin content.
5. Nanoparticle tracking analysis (NTA) is the more widely accepted method for characterizing the particle size of extracellular vesicles/OMVs, supplemented characterizations are needed.
6. The nanoparticle was administrated topically, thus the HMGB1 concentration in tumor tissue should be measured but not in the plasma. (Figure 4e)
7. All previous studies used peritumoral injection, but the intratumoral injection was used in the draining lymph nodes targeting studies. Why would this change? In addition, the mechanism of DLN targeting needs to be further explored.
8. Why the CT26 subcutaneous tumor model was also constructed? which was not in accordance with the pathological characteristics of colon cancer. And why did this model use BALB/c mice instead?
9. The extraction was misspelled as extrusion in Figure 1a.

Reviewer #3 (Remarks to the Author):

Zhuang and colleagues (NCOMMS-22-32734) report on a biochemical, cell biological, and in vivo tumor study to develop and characterize a novel nanosystem strategy for adjuvant cancer immunotherapy that blocks Mertk-mediated efferocytosis to facilitate anti-tumor immunity. Towards this goal, authors first isolate outer membrane vesicles (OMVs) from E Coli strain MG1655, then encapsulate a Mertk small molecule tyrosine kinase inhibitor called UNC2025, and finally derivatize the outer membrane proteins with a maleimide chemical crosslinker called Mal-PEG4-NHS that can capture and covalently crosslink cellular and exogenous proteins. Authors prepare these OMV particles and employ this strategy to inhibit Mertk-mediated efferocytosis and induce secondary necrosis, predicted to release DAMPs and tumor associated antigens (TAA) that can stimulate DC-mediated cross-presentation and anti-tumor immunity. Main features and conclusions in the paper include (i) methodology development to prepare a series of OMV's the prototype being encapsulated UNV-2025 with capture crosslinker (mU@OMV), (ii) assessment that mU@OMV can block efferocytosis and induce secondary necrosis to release TAAs, (iii) shown that mU@OMV's can capture OVA, induce DC maturation and traffic to draining lymph nodes, and (iv) show that mU@OMV's can induce systemic immune responses leading to anti-tumor, anti-metastases, and abscopal effects in B16 and CT26 mouse syngeneic models.

Overall, this is a potentially important and forward-looking paper and strategy, and effects and outcomes seem robust and reproducible. Strengths include the novelty of the approach and the potential universal and agnostic utility of the strategy to stimulate host anti-tumor immunity towards tumor specific neoantigens. While this approach and strategy is meritorious and should be of interest to this research field, several issues came up in the review that should be clarified.

In Fig. 1h, authors show capture of exogenous OVA by mU@OMV but this looks underwhelming. Why are there so few bands in the OMV preparation? Authors should include a silver stain to better articulate the OMV composition.

The data in Fig. 2 is interesting but difficult to reconcile how partial inhibition of Mertk (Fig 2A) can completely and robustly inhibit all efferocytosis (over 90%). Previous studies have shown that PS receptors combine to drive efferocytosis (for example Mertk and TIM4 in macrophages or Mertk and avb5 in DCs). Additional support is needed to show UNC2025 is acting specifically on Mertk and (i) not on other TAMs, or (ii) acting non-specifically on phagocytosis processes.

More detail is required to assess data in Fig. 2g and 2h. What is relative abundance defined as? Relative to U@OMV?

Authors call the mU@OMV effects "necrocytosis"? What does this mean. If they mean immunogenic cell death, they should assess externalized ecto-calreticulin (CRT), and if they mean necroptosis, should assess pMLKL.

In Fig. 3, 12 hours seems too long for DC uptake. Did authors try other time points. In addition, what is the receptor for mU@OMV on the DCs. What are the TAAs?

Likewise, why are not DCs inhibited mU@OMV. Do they not express TAM/Mertk receptors?

The in vivo data in Fig. 5-8 are very nice and a strength in the paper, particularly in 2 models. Additional studies identifying mechanisms of cross presentation will provide important information to this field.

Responses to the comments from the reviewers and editor. (Note: All the changes have been highlighted by giving the text a blue background in the revised manuscript.)

Reviewers' comments:

Reviewer #1

In this manuscript, the authors developed a versatile nanosystem for efficient tumor immunotherapy by synergizing the efferocytosis blockade-induced secondary necrosis with the bacterial outer membrane vesicles (OMVs)-based antigen transfer and immune augment. mU@OMVs effectively inhibited exocytosis, prevented MerTK phosphorylation in tumor-associated macrophages (TAMs), and captured the released tumor-associated antigens (TAAs). Moreover, OMVs are loaded with antigens and then drain to lymph nodes, thereby triggering a strong T-cell immune response and inhibiting tumor progression in multiple tumor models. I am optimistic for this manuscript and recommend that the study should be published after addressing the following major suggestions.

1. The main concern is the novelty of the research. The construction and design of vectors based on OMVs and the concept of "in situ antigen capture" are similar to the published articles (Li et al, Small, 2022).

Li et al (Small, 2022) presented an OMV-based multifunctional in situ vaccine with capacities of antigen capture and immune modulation to enhance immune-mediated tumor clearance after PTT. Briefly, the surface of native OMVs was modified with maleimide groups (Mal) (OMV-Mal) to capture large amount of tumor antigens after PTT, while 1-methyl-tryptophan (1-MT, an inhibitor of indoleamine 2, 3-dioxygenase (IDO)) was loaded in OMV-Mal into OMV-Mal (1-MT@OMV-Mal) by electroporation to cover the immunosuppressive microenvironment mediated by Tregs. In situ injection of 1- MT@OMV-Mal simultaneously overcomes the immune inhibition of IDO on tumor-infiltrating effector T cells, leading to remarkable inhibition on both primary and distant tumors. Please compare the differences and advantages of the two studies in detail in the discussion and add it as reference.

Response: We appreciate the referee's positive comments and kind advice.

The work of Li et al (Small, 2022, 18, e2107461) is very interesting. The innovation of this study was the combination of OMV and 1-MT to solve the two problems related to photothermal therapy (PTT): the limited recognition of the released antigens and the immunosuppressive microenvironment leading to deficient immunity. OMV-Mal captured tumor-released proteins and improved the uptake of

antigens by local DCs, meanwhile, 1-MT overcame the immunosuppressive microenvironment by inhibiting the IDO pathway. The cooperation of PTT and 1-MT@OMV-Mal lead to remarkable tumor inhibition effects.

In our work, an OMVs-based nanosystem is constructed to block the physiological efferocytosis of apoptotic tumor cells by tumor-associated macrophages (TAMs) and transfer the subsequently released tumor antigens to the lymph nodes (LNs) to trigger systemic tumor-specific immunity. Although the surface of OMVs is also modified with Mal to capture antigens, the concept and focus of our work are different from that of Li et al.

Firstly, the antigen release is induced by blocking the ubiquitous efferocytosis, an approach that has not been reported. Tumor antigen is always the critical factor to initiate antitumor immune response. *In situ* antigens with a broad spectrum of whole-antigen epitopes can stimulate powerful immune responses, and thus are recently employed to construct more effective vaccines compared with traditional vaccines prepared *in vitro* (Nat. Commun., 2020, 11, 1110; Adv. Mater., 2018, 30, 1705581). PTT, radiotherapy, photodynamic therapy or chemotherapy was the most common method for *in situ* antigen generation (Nat. Rev. Cancer, 2012, 12, 860; Small, 2022, 18, e2107461). However, tumors often develop resistance to such therapies due to various reasons, e.g. the thermoresistance induced by heat-shock responses, radioresistance caused by the repair response to DNA damage, PDT resistance resulting from antioxidant detoxifying enzymes, and chemoresistance derived from drug efflux (Angew. Chem. Int. Ed., 2021, 60, 12971; Nature, 2006, 444, 756; Chem. Soc. Rev., 2021, 50, 9152; Nat. Commun., 2021, 12, 2804). Moreover, the apoptotic tumor cells induced by these methods will face the risk of efferocytosis by TAMs, reducing the release of TAAs while enhancing the immunosuppressive microenvironment (Cancer Res., 2022, 82, 1991; J. Immunother. Cancer, 2020, 8, e001408). In our study, efferocytosis is inhibited by UNC2025-loaded OMVs. As a result, the apoptotic cells sourced from the widespread cellular stress in tumors will develop into secondary necrosis. Hence, the entire spectrum of TAAs can be released while the immunosuppressive microenvironment is ameliorated.

Secondly, *in situ* TAAs are efficiently hitchhiked to the immune cell-enriched LNs with the aid of mU@OMVs. The effect of *in situ* antigens is inevitably compromised by the dysfunctional and rare DCs in the tumor microenvironment. Without recruiting more DCs or reversing the dysfunctional DCs, the tumor-specific immunity is hard to be activated efficiently in tumors (Nat. Biomed. Eng., 2022, 6,

44; ACS nano, 2019, 13, 3083). Different from the strategies that improve the uptake of antigens by

Figure 4 **f** *Ex vivo* imaging of the DLNs collected 24 h after administration. **g** Quantitative statistics of mean fluorescence intensity (MFI) of Cy5 channel within the region of interest (ROI) of isolated DLNs.

Figure 5 **e** Percentages of MHC-I⁺ cells in CD11c⁺ CD8⁺ DCs and CD11c⁺ CD8⁻ DCs in DLNs. Data in **e** are presented as mean \pm s.d. (n = 3). Statistically significant differences between groups were identified by one-way ANOVA. ****P < 0.0001, ***P < 0.001, **P < 0.01, *P < 0.05.

local DCs, our mU@OMVs system can transfer the *in situ* TAAs out of tumors to immune cell-enriched LNs. The codelivery of massive TAAs and novel adjuvant OMVs to LNs effectively activates the resident CD8⁺ DCs in LNs, and thus induces the effective antitumor immune response (Fig. 4f, g and Fig. 5e).

Therefore, our smart nanosystem is integrated with versatile functions, including antigen initiation, antigen capture, antigen transferring, and adjuvant supplement. This all-in-one system induces intense immune responses and thus efficient tumor inhibition outcomes in different tumor models. Moreover, this strategy is easy to be realized and generally applied.

We have added the differences and advantages of the two studies in detail in the discussion part on page 19 lines 21-28 and added Li's study as reference 30 in the revised manuscript.

2. The gating strategy for the flow cytometry analyses should be provided.

Response: Thanks for this kind suggestion. We have provided all the gating strategies of the flow cytometry analyses in the revised supporting information (Supplementary Fig. 15, 17, 19, 22, 24, 26, 27, and 32, respectively).

3. In addition to T cells, key immunosuppressive cells in tumor tissue should be analyzed. At the same time, the immune microenvironment in the distal tumor should be analyzed simultaneously.

Response: Thanks for these constructive comments. According to the suggestions, both the tumor-infiltrating CD8⁺ cytotoxic T lymphocytes (CTL) and immunosuppressive regulatory T cells (Treg) in

tumor microenvironments have been investigated in the bilateral model. For which, the primary and

abscopal tumors were isolated and analyzed on the 7th day after the final administration of PBS, UNC2025, OMVs, U@OMVs or mU@OMVs.

As shown in Response Figure 2 a-d, both primary and abscopal tumors with PBS treatment showed severe immunosuppression, since the frequency of CD8⁺ CTL was limited (about 1 % in primary site and 3 % in distant site) while the percentage of Treg was high (about 20 % in primary site and 35 % in distant site). UNC2025 alone could not reverse the immunosuppression in either primary or distant tumors. OMVs or U@OMVs treatment could enhance the proportions of CD8⁺ CTL to a certain degree probably owing to the immune modulation ability of OMVs (Adv. Mater., 2020, 32, e2002085). However, the highest frequency of infiltrating CTLs (3.6 %) was observed after mU@OMVs treatment while the frequency of Treg significantly decreased to 7.5 % in the primary tumors. Similarly, the percentage of CD8⁺ T cells in abscopal tumors in mU@OMVs group was almost 2 folds than that of other groups. In contrast, the frequency of Treg in the abscopal tumors significantly decreased to 5.2 %, which was individually 6.8, 6.2, 3.3, or 2.7-fold lower than that of PBS, UNC2025, OMVs, or U@OMVs group. Therefore, a significant increase of CD8⁺ T/Treg ratio (P-values < 0.0001) was observed in the abscopal tumors, which were individually 14.7, 16.6, 5.2, and 4.7-fold higher than that of PBS, UNC2025, OMVs, and U@OMVs groups, demonstrating that mU@OMVs could effectively activate the antitumor immune response meanwhile regulate the immunosuppressive microenvironment. We have revised the corresponding explanation in the revised manuscript on page 16 line 15 to line 22 and added the related results in Fig. 6g-i and Supplementary Fig. 27.

4. When OMVs capture tumor antigen, it is often taken up by local APCs and transported to draining lymph nodes in a cell-dependent manner. In this study, this process was not characterized. In addition, does inhibition of macrophage efferocytosis affect their ability to take up OMVs and antigen?

Response: Thanks for this constructive comment and inquiry. Indeed, local APCs will take up some of the antigen-captured OMVs. But local APCs, which are usually referred to as migratory DCs, are few in tumor sites, and only a small fraction of the antigens can be directly taken up by migratory DCs (Nat. Biomed. Eng., 2022, 6, 44; Adv. Mater., 2022, 32, e2002085). Thus, antigens delivered by local APCs are very limited. In this work, we designed the mU@OMVs system that could effectively migrate from tumors to the immune cell-enriched LNs (Fig. 4f, g). Hence, abundant *in situ* TAAs could be transferred to LNs.

CD8⁺ DCs mainly reside in LNs and are more efficient in cross-presenting tumor antigens for cytotoxic T lymphocyte activation (J. Exp. Med., 2009, 206, 359; Proc. Natl. Acad. Sci., 2016, 113, 1044). To

further prove the potency of mU@OMVs in delivering antigens to resident DCs in LNs, we have comparatively investigated the expression of major histocompatibility complex I (MHC-I) in CD8⁺ DCs in the draining lymph nodes (DLNs) after different treatments (Response Figure 3). Briefly, B16F10 tumor-bearing C57BL/6 mice were peritumorally injected with PBS, UNC2025, OMVs, U@OMVs, or mU@OMVs. Then, DLNs were extracted at 48 h post-administration and the proportion of MHC-I in

CD11c⁺ CD8⁺ DCs was measured by flow cytometry. As expected, MHC-I expression in CD8⁺ DCs was significantly enhanced after mU@OMVs immunization, indicating its satisfactory antigen cross-

presentation in CD8⁺ DCs. Taken together, these results indicated that antigens captured by mU@OMVs could be transferred out of tumors to immune cell-enriched LNs, wherein, the resident CD8⁺ DCs were activated, leading to subsequent CD8⁺ T cell stimulation and antitumor effects. We have revised the corresponding explanation in the revised manuscript on page 12 lines 28-30 and page 14 lines 1-4 and added these results in Fig. 5e and Supplementary Fig. 22.

As shown in Supplementary Fig. 7, U@OMVs and mU@OMVs can also be effectively phagocytosed by M2 macrophages as native OMVs. This was reasonable because the efferocytosis and endocytosis of macrophages are mediated by different receptors. Tyrosine kinase MerTK contributes to efferocytosis and clathrin-mediated endocytosis is the internalization pathway of OMVs (Cell Metab., 2021, 33, 2445; Cell, 2016, 165, 1106). Therefore, the inhibition of macrophage efferocytosis will not compromise their uptake of OMVs and antigens.

5. In figure 1h, in addition to the OVA bands, there were some different protein bands between OMVs and group 3-5. Please explain.

Response: We appreciate this kind inquiry. In Fig. 1h, samples 1 and 2 are OVA and OMVs, respectively. Samples 3-5 were obtained by the following steps. Firstly, the same amount of OMVs, U@OMVs and mU@OMVs were individually incubated with OVA. Then, the unabsorbed OVA was separated from these vesicles by ultracentrifugation. And the obtained OVA-absorbed vesicles (OMVs+OVA, U@OMVs+OVA and mU@OMVs+OVA) were used for SDS-PAGE analysis.

We are sorry for the confusing results in figure 1h. We here provide the original data of SDS-PAGE to make sure the distribution of protein bands (Response Figure 4). As could be seen, the protein bands in groups 3-5 belonged to those of OVA or OMVs. Compared with proteins from OMVs (indicated by yellow arrows), the protein bands corresponding to OVA (between 35-48 kd, indicated by blue arrows) were obviously found in mU@OMVs+OVA group (sample 5). Although these bands were also visible in OMVs+OVA and U@OMVs+OVA groups (samples 3, 4), they were much weaker than those in mU@OMVs+OVA group. We have added the indicating arrows in Fig. 1 h in the manuscript for better comparison.

6. In figure 3j, CD69 is a non-specific activation marker of T cells. In addition, the IFN- γ ⁺ CD8⁺ T cells are also not antigen-specific T cells. Please add antigen specific experiments to prove the antigen capture of mU@OMVs. For example, the IFN- γ ⁺ CD8⁺ T cells should be detected after antigen re-challenge using ELISPOT.

Response: Thanks for the referee's careful review and insightful comments. It is true that CD69 often acts as a costimulatory molecule for early T cell activation but not the specific activation marker for T

cell (Nature, 2021, 592, 290), and IFN- γ ⁺ CD8⁺ T cells can destroy tumor cells powerfully but are also not definite antigen-specific T cells (Nat. Cancer, 2020, 1, 749). To better verify the activation of antigen-specific immune responses, T cells are often stimulated with antigen *in vitro*, and then the secretion of cytokines like IFN- γ is measured by ELISpot (enzyme-linked immunosorbent spot) assay, flow cytometry, ELISA, and so on (Nat. Protoc., 2009, 4, 46). According to the suggestion, we have analyzed the antigen-specific immune response. Briefly, B16F10 tumor-bearing C57BL/6 mice were peritumorally injected with PBS, UNC2025, OMVs, U@OMVs, or mU@OMVs (Response Figure 5). On day 25, the splenocytes from the immunized mice were collected and restimulated *ex vivo* with B16F10 antigen for 24 h. Next, the expression of IFN- γ in CD3⁺ CD8⁺ T cells was measured by flow cytometry. The results showed that the proportion of IFN- γ ⁺ T cells in mU@OMVs group was increased by 27.8 %, 24.2 %, 16.9 % or 13.1 % compared with that in PBS, UNC2025, OMVs, or U@OMVs group, respectively, suggesting that mU@OMVs could elicit a strong antigen-specific immune response. We have revised the corresponding explanation in the revised manuscript on page 14 line 9 to line 15 and added the related results in Fig. 5j, k and Supplementary Fig. 24.

Reviewer #2

In this manuscript, the authors construct a nanosystem composed of bacterial outer membrane vesicles loaded with UNC2025 and modified with maleimide on the surface. This nanosystem can not only inhibit efferocytosis but also boost the following antitumor immunity, which efficiently prevents the growth, metastasis, and recurrence of tumors in mice.

Here are some issues that need to be explained or revised about this project:

1. Why the nanoparticles were administrated peritumorally, but not intravenously or intratumorally?

Response: Thanks for the referee's valuable comments and kind inquiries. Compared with systemic administration, local administration, particularly subcutaneous injection, is popular in cancer vaccination since it can increase the concentration and promote penetration of drugs in tumors, and thus enhanced treatment outcomes with reduced systemic toxicity could be achieved (Nat. Rev. Clin. Oncol., 2021, 18, 558). Moreover, local tumor immunotherapy has been proved to be able to enhance the infiltration of immune cells and initiate a more powerful systemic immune response, thus effectively overcoming both primary tumors and metastases (J. Control. Release, 2021, 329, 882).

However, intratumoral injection, another kind of local administration, may lead to tumor hemorrhage and the needle track may spread tumor cells to induce metastases (Adv. Mater., 2022, 34, e2206915; Oncoimmunology, 2019, 8, e1625687). More importantly, melanoma in this manuscript is aggressive cancer (Nat. Rev. Cancer, 2022, 22, 195), and the intratumoral accessibility of melanoma would face a higher risk of metastases. Therefore, we chose the peritumoral administration of mU@OMVs here.

2. As originated from the biomembrane, OMVs are rich in thiol, which makes it unreasonable to use DSPE-PEG-Mal for surface modification. How to ensure and characterize that DSPE-PEG-Mal is inserted into the membrane through hydrophobic interaction, rather than the covalent bond between thiol and Mal? Besides, even if Mal can be modified on the surface of OMVs by the pattern described in this manuscript. I am afraid the OMVs would be cross-linked, therefore triggering the dramatic nanoparticle size increase observed in Figure 1.

Response: We appreciate these kind inquiries.

In this work, Mal was introduced onto the surface of OMVs through a bioconjugate reaction between Mal-PEG4-NHS and amino groups on the bacterial outer membranes (described in the manuscript on page 4 lines 27-28). Therefore, Mal-PEG4-NHS was modified on the membrane through covalent

	OMVs	U@OMVs	mU@OMVs
Concentration (Particles/mL)	5.9×10^8	5.3×10^8	5.5×10^8
Relative change	100 %	89.83 %	93.22 %

Response Table 1 The concentration (particles/mL) of OMVs, U@OMVs or mU@OMVs.

Fig. 1 g Stability of mU@OMVs in PBS at 4 °C detected by DLS.

amide crosslinks, rather than hydrophobic interaction. We are sorry for this confusion.

NHS usually reacts quickly with the primary amino at neutral to basic pH values (pH 7.2-9.0) (Methods Enzymol., 2014, 536, 87). Whereas Mal reacts with thiol to form stable thioether bonds at neutral to acid pH values (6.5-7.5) and preferably as low as possible within that range since the specificity of this reaction is much higher at acid pH (Bioconjugate Techniques, 2013, 229; Methods Enzymol., 2014, 536, 79; J. Protein Chem., 1983, 2, 263). According to the different pH-dependent manners of the two reactions, U@OMVs were firstly left to react with Mal-PEG4-NHS for 3 h to obtain mU@OMVs. This reaction was performed in physiological pH 7.4 PBS and therefore, the Mal-PEG4-NHS would preferentially react with amino groups on OMVs via the NHS functional end, leaving the Mal free as illustrated in the instructions of Mal-PEG4-NHS (Nat. Nanotechnol., 2012, 12, 877; Adv. Mater., 2019, 31, e1902626; Acta Biomater., 2020, 101, 422). In the subsequent *in vitro* antigen capture test, the pH was adjusted to 6.8, in which condition the reaction of Mal with thiol proceeds at a rate more than 1000 times greater than its reaction with amino groups (Bioconjugate Techniques, 2013, 241), and thus efficient crosslinking between Mal on OMVs and thiol on antigens could be realized (described in the manuscript on pages 22 and 24). After mU@OMVs were injected peritumorally, the acidic tumor microenvironment would be beneficial for the antigen capture by mU@OMVs (Nat. Commun., 2022, 13, 2336).

To further address the concern about mU@OMVs crosslinking, nanoparticle tracking analysis (NTA) was used to measure the concentrations of OMVs, U@OMVs and mU@OMVs (no purification steps to prevent the loss, but standing overnight at 4 °C in pH 7.4 PBS). As shown in Response Table 1, the concentrations of U@OMVs and mU@OMVs were maintained at nearly 90 % compared with that of OMVs. Moreover, our size stability study also showed that the size of mU@OMVs was almost kept during 7 days incubation (Fig. 1g), confirming the negligible crosslinking in mU@OMVs. We have revised the corresponding explanation in the revised manuscript on page 5 lines 4-10 and added the related results in Supplementary Table 1 in the revised supporting information.

3. Why did the particle size increase so much after being encapsulated with UNC2025?

Response: Thanks for this constructive comment. The particle size displayed in Fig. 1e-g was determined by the dynamic laser scattering (DLS) and the hydrodynamic diameters of OMVs, U@OMVs and mU@OMVs were respectively 38.69 ± 4.84 nm, 77.09 ± 14.09 and 100.04 ± 1.70 nm, similar to those in references (ACS nano, 2014, 8, 1525; ACS nano, 2021, 15, 13826). The size increase was probably because DLS was biased towards large particles or aggregation in its statistical data analysis (Langmuir, 2015, 31, 3; Int. J. Pharm., 2020, 573, 118802), and UNC2025 was loaded into OMVs through electroporation, which occasionally made the fusion or aggregation of vesicles (J. Control. Release., 2013, 172, 229; J. Extracell. Vesicles, 2019, 8, 1650595). Actually, we studied the size change of OMVs in different electroporation conditions. As shown in Response Figure 6, the DLS size of OMVs gradually increased with the increasing of voltage and drastically reached more than 1 μ m at the voltage of 300 v. According to our results, 100 v was used to electroporate UNC2025 into OMVs due to the much less influence on size. Even if at the optimized electroporation condition, some U@OMVs with large size may be present. Despite very low proportions, they may lead to a clear size increase in DLS results.

To further assess the sizes of different OMV formations, nanoparticle tracking analysis (NTA), one more widely accepted method for characterizing exosomes or vesicles (J. Extracell. Vesicles., 2013, 2, 19671), was also employed here. As could be seen, there were no obvious changes in the size distribution of OMVs, U@OMVs or mU@OMVs (Response Figure 7). The possible reason is that although both DLS and NTA determine particle size by measuring the diffusion coefficient, DLS reads the intensity change of scattered light to find the diffusion coefficient of particles, while NTA calculates

Response Figure 6 Size of OMVs under different electroperoration conditions. Data are presented as mean \pm s.d. (n = 6).

Response Figure 7 Nanoparticle tracking analysis (NTA) of OMVs, U@OMVs and mU@OMVs.

the diffusion coefficient based on the movements of individual particles in optical video images (J. Extracell. Vesicles, 2013, 2, 19671; J. Extracell. Vesicles, 2017, 6, 1333883). Therefore, DLS biases towards large particles or aggregation in its statistical data analysis, while the results of NTA measurement would not be deflected if the large particles in samples are very few (Methods Mol. Biol., 2017, 1494, 239). We have provided the NTA results in Supplementary Fig. 5 in revised supporting information to illustrate the size distribution of OMVs, U@OMVs or mU@OMVs from different aspects.

4. As a potentially pathogenic bacterium, is the *E. coli* used for the extraction attenuated? If not, more safety evaluations are needed, such as the investigation of endotoxin content.

Response: We appreciate the referee's instructive comments. OMVs-based 4CMenB vaccines against invasive Serogroup B *Neisseria meningitidis* have been approved by the European Commission, indicating that OMVs can be safely employed (NPJ vaccines, 2021, 6, 130). In our work, OMVs were extracted from the *E. coli* MG1655, a non-pathogenic strain for laboratory application (Nat. Commun., 2018, 9, 1680; Dis. Model. Mech., 2018, 11, dmm035063). Our results showed that the biomarkers of hepatic function, renal function, and cardiac function were all normal in mU@OMVs-

Supplementary Figure 29 Serum biochemical indexes of ALP (a), ALT (b), AST (c), CREA (d), BUN (e) and LDH (f) in different groups at the endpoint of the antitumor experiment against B16F10.

Supplementary Figure 30 H&E staining of major organs at the endpoint of the antitumor experiment against B16F10.

treated mice (Supplementary Fig. 29a-f), and H&E staining of tissues also showed no obvious damage after treatment (Supplementary Fig. 30), demonstrating the satisfactory safety profile of our mU@OMVs-based antitumor therapy.

Response Figure 8 Quantification of endotoxin in different types of OMVs. The content of endotoxin was detected by chromogenic LAL endotoxin assay kit. Data are presented as mean \pm s.d. (n = 6). Statistically significant differences between groups were identified by one-way ANOVA. ****P < 0.0001, ***P < 0.001, **P < 0.01, *P < 0.05.

Response Figure 9 *In vivo* systemic safety of OMVs after intravenous injection (O-10: 10 μ g OMVs; O-20: 20 μ g OMVs). Data are presented as mean \pm s.d. (n = 3).

To further address the concerns from the reviewer, we also comparatively evaluated the endotoxin content on OMVs of three different *E. coli* strains. As shown in Response Figure 8, the endotoxin content on OMVs derived from *E. coli* MG1655 was individually 3.6 and 2.5-fold lower than that of BL21 or DH5 α strain with the same protein amount. For the same number of OMVs, the endotoxin in MG1655 group was also significantly less than the other two groups, confirming the low toxicity of OMVs derived from *E. coli* MG1655. The possible reason was that *E. coli* MG1655 had the shorter LPS length since the expression of O-antigen was usually removed in the laboratory *E. coli* K-12 strains including *E. coli* MG1655 (Nat. Commun., 2022, 13, 6195).

Also, we have tested the *in vivo* safety of OMVs derived from *E. coli* MG1655 and provided the related results here (Response Figure 9). In brief, mice were individually injected (i.v.) with PBS, 10 ug or 20 ug OMVs, and the sera were collected at different time points to detect the biochemical index levels. The results showed that the secretion levels of alanine aminotransferase (ALT), aspartate transaminase (AST), blood urea nitrogen (BUN) and lactate dehydrogenase (LDH) were all within normal ranges within 72 h. Together, we believed that OMVs used in our study had a satisfactory biosafety profile.

5. Nanoparticle tracking analysis (NTA) is the more widely accepted method for characterizing the particle size of extracellular vesicles/OMVs, supplemented characterizations are needed.

Response: We appreciate the valuable suggestions from the reviewer. NTA is indeed the more widely accepted method for characterizing the particle size of extracellular vesicles because NTA can track the Brownian motion of individual vesicles by real-time visualization and can calculate the size and total concentration of vesicles (Anal. Chem., 2019, 91, 15, 9508). Our NTA results revealed the uniform size distributions of all three vesicles (Response Figure 10). We have revised the corresponding explanation in the revised manuscript on page 5 lines 4-7 and the corresponding results have been provided in Supplementary Fig. 5 in the revised supporting information.

6. The nanoparticle was administrated topically, thus the HMGB1 concentration in tumor tissue should be measured but not in the plasma. (Figure 4e)

Response: Thanks for the reviewer's kind comment. Dying tumor cells can generate a large amount of HMGB1 in the extracellular environment, leading to the release of HMGB1 into the blood. So, the

Response Figure 11 HMGB1 levels in tumor tissues after different treatments determined by ELISA. Data are presented as mean \pm s.d. (n = 6). Statistically significant differences between groups were identified by one-way ANOVA. ****P < 0.0001, ***P < 0.001, **P < 0.01, *P < 0.05.

Fig. 4 e Serum HMGB1 levels of mice after different treatments determined by ELISA.

analysis of HMGB1 concentration in plasma was widely used in research and in clinic probably owing to its convenience (Nat. Biomed. Eng., 2020, 4, 1102; Cell, 2010, 140, 798; Cell Death Dis., 2018, 9, 648). Also, HMGB1 concentration in tumor tissues could be measured to evaluate the therapeutic effects. According to the suggestion, we have measured the HMGB1 concentration in tumor tissues at 48 h after the last administration. As shown in Response Figure 11, the levels of HMGB1 in mU@OMVs treated mice were significantly enhanced than those in other groups, consistent well with the results in Fig. 4e. We have revised the corresponding explanation in the revised manuscript on page 10 lines 29-30 and page 11 line 28 and corresponding results have been added as Supplementary Fig. 18 in the revised supporting information.

7. All previous studies used peritumoral injection, but the intratumoral injection was used in the draining lymph nodes targeting studies. Why would this change? In addition, the mechanism of DLN targeting needs to be further explored.

Response: Thanks for this careful review and we are sorry for this confusion. As a matter of fact, we evaluated both the peritumoral injection and intratumoral injection in the LNs targeting studies but only the intratumoral injection results were presented in the manuscript (Fig. 4f, g). Here, we also provided

Figure 4 f Ex vivo imaging of the DLNs collected 24 h after intratumoral injection. **g** Quantitative statistics of mean fluorescence intensity (MFI) of Cy5 channel within the region of interest (ROI) of isolated DLNs.

Response Figure 12 a Ex vivo imaging of the DLNs collected 24 h after peritumoral administration. Briefly, Cy5-labeled OVA were incubated with OMVs, U@OMVs or mU@OMVs for 3 h at 37 °C and then peritumorally administrated to B16F10 melanoma-bearing mice, and the DLNs were collected 24 h after injection. Then, OVA signals in the isolated DLNs were analyzed by the *in vivo* imaging system (IVIS). **b** Quantitative statistics of mean fluorescence intensity (MFI) of Cy5 channel within the region of interest (ROI) of isolated DLNs. Data in a, b are presented as mean ± s.d. (n = 3). Statistically significant differences between groups were identified by one-way ANOVA. ****P < 0.0001, ***P < 0.001, **P < 0.01, *P < 0.05.

the peritumoral injection results. As shown in Response Figure 12, free OVA, OVA in OVA+OMVs or OVA+U@OMVs group could migrate to DLNs to a certain extent after peritumoral injection. As expected, the accumulation of OVA in DLNs in the OVA+mU@OMVs group was significantly higher than those in other groups, similar to the intratumoral injection results in Fig. 4f, g.

Both results demonstrated that mU@OMVs was capable of transferring TAAs to DLNs. The reason for displaying the results of intratumoral injection in the manuscript was that tumor antigens are difficult to cross the tumor stroma owing to the dense extracellular matrix in the tumors (Nat. Biomed. Eng., 2022, 6, 44). Therefore, intratumoral injection of mU@OMVs was closer to the actual situation of transporting tumor antigens out of tumors towards LNs. Indeed, our results showed that compared with peritumoral injection, much fewer antigens could be transferred to LNs by mU@OMVs after intratumoral injection, and the mean fluorescence intensity (MFI) of the peritumoral group was about 1.6-1.9 times that of the intratumoral group.

Generally speaking, lymphatic delivery systems can drain to LNs along with the interstitial flow (passive targeting), and the targeting efficacy greatly lies in the size and surface features of the systems. Typically, small particles with size between 20 to 200 nm are able to drain freely to the LNs (Nat. Commun., 2017, 8, 1954; ACS nano, 2016, 10, 2678). OMVs can efficiently drain into LNs due to the extremely appropriate size (Biotechnol. Adv., 201, 35, 565). Moreover, the pathogen-associated molecular patterns (PAMPs) on OMVs surface facilitate their retention in LNs through the specific recognition and then internalization of DCs (Adv. Mater., 2020, 32, e1908185). We have added the explanation about the LNs targeting ability of OMVs in the revised manuscript on page 12 lines 1-5.

8. Why the CT26 subcutaneous tumor model was also constructed? which was not in accordance with the pathological characteristics of colon cancer. And why did this model use BALB/c mice instead?

Response: We appreciate these kind inquiries. Different tumor models exhibit distinct characteristics and responses to immunotherapy, closely related to the tumor immunogenicity across tumor types, tumor immune microenvironment, tumor immune cell infiltration, effector T cell function and so on (Nat. Rev. Cancer, 2019, 19, 215; Nat. Med., 2022, 28, 1421; Adv. Mater., 2022, 34, e2106307). Hence, besides the melanoma model, we also constructed a CT26 tumor model in this study to comprehensively evaluate the antitumor immune effects of mU@OMVs. Compared with the orthotopic transplantation model, the subcutaneous tumor model is more efficient and cost-effective, hence they

have been extensively employed for cancer drug development (Nat. Protoc., 2022, 17, 2108). It is true that the orthotopic transplantation model would be more consistent with the pathological characteristics of colon cancer. But as we approved in the manuscript, mU@OMVs could induce systemic antitumor immune responses, and thus the subcutaneous tumor model could imitate the therapeutic effects of mU@OMVs on orthotopic model to a certain extent.

CT26 is an N-nitroso-N-methylurethane-(NNMU) induced undifferentiated colon carcinoma cell line established from BALB/c mice with aggressive colon carcinoma. Therefore, BALB/c mice were used here for building syngeneic tumor models (J. Biomed. Phys. Eng., 2021, 11, 281). We are sorry for this confusion.

9.The extraction was misspelled as extrusion in Figure 1a.

Response: Thanks very much for the careful review and we are so sorry for this mistake. It has been revised in the manuscript.

Reviewer #3

Zhuang and colleagues (NCOMMS-22-32734) report on a biochemical, cell biological, and in vivo tumor study to develop and characterize a novel nanosystem strategy for adjuvant cancer immunotherapy that blocks Mertk-mediated efferocytosis to facilitate antitumor immunity. Towards this goal, authors first isolate outer membrane vesicles (OMVs) from E Coli strain MG1655, then encapsulate a Mertk small molecule tyrosine kinase inhibitor called UNC2025, and finally derivatize the outer membrane proteins with a maleimide chemical crosslinker called Mal-PEG4-NHS that can capture and covalently crosslink cellular and exogenous proteins. Authors prepare these OMV particles and employ this strategy to inhibit Mertk-mediated efferocytosis and induce secondary necrosis, predicted to release DAMPs and tumor associated antigens (TAA) that can stimulate DC-mediated cross-presentation and antitumor immunity. Main features and conclusions in the paper include (i) methodology development to prepare a series of OMV's the prototype being encapsulated UNV-2025 with capture crosslinker (mU@OMV), (ii) assessment that mU@OMV can block efferocytosis and induce secondary necrosis to release TAAs, (iii) shown that mU@OMV's can capture OVA, induce DC maturation and traffic to draining lymph nodes, and (iv) show that mU@OMV's can induce systemic immune responses leading to antitumor, anti-metastases, and abscopal effects in B16 and CT26 mouse syngeneic models.

Overall, this is a potentially important and forward-looking paper and strategy, and effects and outcomes seem robust and reproducible. Strengths include the novelty of the approach and the potential universal and agnostic utility of the strategy to stimulate host antitumor immunity towards tumor specific neoantigens. While this approach and strategy is meritorious and should be of interest to this research field, several issues came up in the review that should be clarified.

1. In Fig. 1h, authors show capture of exogenous OVA by mU@OMV but this looks underwhelming. Why are there so few bands in the OMV preparation? Authors should include a silver stain to better articulate the OMV composition.

Response: Thank the reviewer for the positive comments on our manuscript. We appreciate the referee's kind inquiry and instructive suggestion. The reasons for the few bands in OMVs were that we only displayed the bands between 35-63 kd, and the loading amounts of proteins are relatively low (about 1 µg). As suggested, the silver stain was used here to better articulate the OMVs compositions. As shown in Response Figure 13, OMVs, U@OMVs as well as mU@OMVs displayed the similar

protein distribution, again proving that electroporation and surface modification did not affect the protein components in OMVs. In addition, the major proteins in OMVs were mainly distributed in 35 kd (may represent outer membrane protein), 48-63 kd (may represent flagellin protein), and 75-100 kd, consistent with the coomassie brilliant blue staining results in Fig. 1h.

We also compared the SDS-PAGE protein analysis results of OMVs from different references. As a whole, OMVs, derived from the laboratory *Escherichia coli*, including MG1655 we used here, are usually simple in protein compositions (Nano Lett., 2021, 21, 8609; Small, 2022, 18, e2107461). However, OMVs from *Salmonella* usually have complex compositions, mainly due to the specific proteins such as SipC, SopE2, SEN4247 in their outer membranes (Adv. Mater., 2020, 32, e1908185; Nano Lett., 2022, 20, 11). We have revised the corresponding explanation in the revised manuscript on page 5 lines 3-4 and the related results were added as Supplementary Fig. 4 in the supporting information.

2. The data in Fig. 2 is interesting but difficult to reconcile how partial inhibition of Mertk (Fig 2A) can completely and robustly inhibit all efferocytosis (over 90%). Previous studies have shown that PS receptors combine to drive efferocytosis (for example Mertk and TIM4 in macrophages or Mertk and avb5 in DCs). Additional support is needed to show UNC2025 is acting specifically on Mertk and (i) not on other TAMs, or (ii) acting non-specifically on phagocytosis processes.

Response: Thanks for the referee's careful review and insightful comments.

Efferocytosis is mediated by the recognition between "eat-me" signals like PtdSer on apoptotic cells and the recognition receptors on phagocytes. Actually, the recognition of PtdSer by phagocytes can not only directly through the receptors including the T-cell immunoglobulin mucin (TIM) family, but also indirectly *via* TAM receptors, including Tyro3, Axl and MerTK (Cell Commun. Signal, 2020, 18, 71).

Emerging studies revealed that targeting either TIM or TAM receptors on phagocytes is highly efficient in inhibiting the engulfment of apoptotic cells (> 90%) like our results in Fig.2 because the PtdSer receptors are not severally driving efferocytosis but in an intricate collaborative array (Immunity, 2020, 52, 357; Cell Death Dis., 2021, 12, 538). Although many PS receptors are studied as individual receptors, multiple PS receptors may be activated at the same time, inducing complex function cooperation and signaling synergy (Immunol. Rev., 2017, 280, 149). For example, the engulfment of apoptotic cells by phagocytes through Tim4 is realized in two steps: at first, Tim4 directly binds to the receptor on apoptotic cells, and then the recruited apoptotic cells are transferred to MerTK for engulfment. It means that MerTK is essential for Tim4-induced efferocytosis. The reason is that TIM receptors lack an intracellular domain and the phagocytosis of apoptotic cells by Tim-4 needs the cooperation of transmembrane receptor MerTK (Cells, 2020, 9, 1625; Mol. Cell Biol., 2014, 34, 1512).

Supplementary Figure 9 Western blot analysis of the phosphorylation of MerTK in M2 macrophages after the treatment with different concentrations of free UNC2025 for 2 h.

Response Figure 14 Western blot analysis of phosphorylation of Axl in M2 macrophages after treatment with different concentrations of free UNC2025 (a) or mU@OMVs (b) for 2 h.

Therefore, although the blockade of MerTK would not suppress other receptors, the phagocytosis behavior induced by other receptors may be remarkably affected, particularly TIM receptors. This may be the reason for the effective efferocytosis inhibition results in Fig.2.

The small molecule inhibitor UNC2025 exhibited much better selectivity on MerTK than other TAMs (e.g. almost 50-fold higher than Axl and 100-fold than Tyro3) (J. Med. Chem., 2014, 57, 7031). As illustrated in Supplementary Figure 9, UNC2025 effectively restrained the kinase activity of MerTK by preventing its phosphorylation in a dose-dependent manner. To further confirm the selectivity of UNC2025 on MerTK, we have tested the kinase activity of another TAM receptor, Axl, in macrophages after free UNC2025 or mU@OMVs treatment. As shown in Response Figure 14, neither UNC2025 nor mU@OMVs treatment decreased the phosphorylation of Axl (p-Axl) in RAW264.7 cells even with high concentrations. We have revised the corresponding explanation in the revised manuscript on page 5 lines 24-27 and the related results were added as Supplementary Fig. 11 in the revised supporting information.

3. More detail is required to assess data in Fig. 2g and 2h. What is relative abundance defined as? Relative to U@OMV?

Response: We appreciate the valuable suggestion from the reviewer. Among all proteins captured by mU@OMV, the intensity of peptide complement C3 (C3) was the highest. Therefore, the relative abundance of proteins captured by mU@OMV was calculated by dividing the values of each peptide intensity by the intensity of peptide C3, meanwhile, the value of the relative abundance of peptide C3 was defined as 10. We have revised the corresponding explanation in the revised manuscript on page 24 lines 6-9 and added this table as Supplementary Table 2 in the revised supporting information.

	proteins	Intensity	Relative abundance		proteins	Intensity	Relative abundance
1	C3	680470000	10	23	Hist1h1c	76945000	1.13076256
2	Pdcd11	473620000	6.96018928	24	Hspa1a	73670000	1.08263406
3	Camk2d	415430000	6.10504504	25	Hmgb1	56022900	0.82329713
4	Eef1a1	323270000	4.75068703	26	Hsp90b1	27793000	0.40843829
5	Actg1	284760000	4.18475465	27	Hspa4	9282000	0.13640572
6	Acbd3	237220000	3.48611989	28	Brd2	455920000	6.70007495
7	Mcm2	160920000	2.36483607	29	Sept7	313240000	4.6032889
8	Flna	155970000	2.29209223	30	Smarcd1	99552000	1.46298882
9	Msn	125410000	1.84299087	31	Trip12	53341000	0.78388467
10	Plaa	99984000	1.46933737	32	Cd34	19807000	0.29107823
11	Dpysl2	79022000	1.16128558	33	Fn1	17335000	0.25475039
12	Larp1	71600000	1.05221391	34	Snx5	10005000	0.14703073
13	Gapvd1	68265000	1.00320367	35	Tubb3	367230000	5.3967111
14	Ehd2	58500000	0.85969991	36	Eef2	268740000	3.94932914
15	Dffb	58415000	0.85845078	37	Actn4	42731000	0.62796303
16	Lrp1	29595000	0.43491998	38	Smarcd1	20781000	0.30539186
17	Wdr1	22698000	0.33356357	39	Aldh18a1	15015000	0.22065631
18	Ctnnd1	12596000	0.18510735	40	Cad	12713000	0.18682675
19	Hspa8	386240000	5.67607683	41	Zc3h14	14202000	0.20870869
20	Hsp90ab1	288320000	4.23707144	42	Ddx27	6022900	0.08851088
21	H2afz	116270000	1.70867195	43	Dag1	2573900	0.03782533
22	Hsp90aa1	107740000	1.58331741				

Response Table 2 The relative abundance of proteins captured by mU@OMVs.

4. Authors call the mU@OMV effects “necrocytosis”? What does this mean. If they mean immunogenic cell death, they should assess externalized ecto-calreticulin (CRT), and if they mean necroptosis, should assess pMLKL.

Response: Thank you for the kind comments. In this work, UNC2025-loaded OMVs are easily recognized and phagocytized by the tumor-associated macrophages (TAMs), resulting in the efficient inhibition of MerTK and thus the efferocytosis blockade of TAMs, finally inducing the secondary necrosis of apoptotic tumor cells. Therefore, we focused on investigating the secondary necrosis (also called necrosis for short sometimes) in this manuscript. We are so sorry for this confusion and have corrected “necrocytosis” to “necrosis” in the revised manuscript on page 7 lines 11-12.

To assess the second necrosis of tumor cells, the concentrations of nuclear high mobility group 1 (HMGB1) proteins, the main signals of necrosis, in both the plasma and the tumor tissues have been

Fig. 4 e Serum HMGB1 levels of mice after different treatments determined by ELISA.

Response Figure 15 HMGB1 levels in tumor tissues after different treatments determined by ELISA. Data are presented as mean \pm s.d. (n = 6). Statistically significant differences between groups were identified by one-way ANOVA. ****P < 0.0001, ***P < 0.001, **P < 0.01, *P < 0.05.

measured. As could be seen from Fig. 4e and Response Figure 15, the levels of HMGB1 in mU@OMVs treated mice were significantly enhanced than those in other groups, demonstrating the massive secondary necrosis of tumor cells. Response Figure 15 has been added as Supplementary Fig. 18 in the revised supporting information.

5. In Fig. 3, 12 hours seems too long for DC uptake. Did authors try other time points. In addition, what is the receptor for mU@OMV on the DCs. What are the TAAs?

Response: Thanks for the referee's careful review and insightful comment. Endocytosis is a cellular process of all eukaryotic cells, including DCs, to ingest macromolecules and particles. The process of endocytosis involves several steps including recognition, internalization, endosome formation and so on (Adv. Drug Deliv. Rev., 2020, 157, 118; Br. J. Cancer, 2021, 124, 66; J. Control. Release, 2010,

145, 182). It is indeed that the initial recognition and internalization of foreign materials by DCs are very fast and can be finished within three hours (Adv. Mater., 2022, 34, e2109984; Sci. Transl. Med., 2021, 13, eabc2816). However, a longer time is often needed to obtain enough ingestion, especially for those molecules that lack corresponding recognition receptors on DCs. For example, in Patel's study, DCs were incubated with free FITC-OVA for 12 h for sufficient uptake (Adv. Mater., 2019, 31, e1902626). In another research, DCs were incubated with OVA for approximate 20 h before flow cytometric analysis (Nat. Biomed. Eng., 2022, 6, 44). The purpose of Fig. 3a was to comparatively study the capacity of mU@OMVs in delivering OVA to BMDCs, thus we chose a relatively long time, 12 h, to observe the cumulative accumulation of OVA in BMDCs. As shown in Fig. 3a and Response Figure 16, only half of BMDCs (52.3 %) uptook free OVA even after 12 h incubation, suggesting that OVA itself could not be effectively ingested by BMDCs. In contrast, almost all BMDCs (94.1 %) were OVA positive and abundant OVA could be seen in BMDCs in OVA+mU@OMVs group owing to the receptor recognition promoted ingestion. The related results were also added as Supplementary Fig. 14 in the supporting information.

Pattern recognition receptors (PRRs) are expressed on a wide class of innate immune cells including DCs, and pathogen-associated molecular patterns (PAMPs) are strong activators of PRRs (Nature immunology, 2001, 2, 675). Therefore, PRRs on DCs are the receptors for recognition of mU@OMVs since multiple PAMPs, such as LPS, lipoproteins, double-strand DNA, and ribosomal RNA, were expressed on OMVs (Adv. Mater. 2020, 32, 2002085; ACS Nano 2021, 12, 1c05613).

Tumor-Associated Antigens (TAAs) refer to the antigen molecules overexpressed on tumor cells, including embryonic proteins, glycoprotein antigens, squamous cell antigens, and so on (Nature, 2021, 590, 157; Semin. Cancer Biol., 2014, 29, 13; J. Am. Chem. Soc., 2013, 135, 14462; Proc. Natl. Acad.

Sci., 1995, 92, 3147). TAAs have been widely used for treating tumors because they can be recognized by antigen-presenting cells to activate tumor-specific T cells and establish adaptive immunity (Nat. Protoc., 2018, 13, 335). Promoting the release of tumor antigens *in situ* can provide a broad spectrum of TAAs and avoid the costly and time-consuming *in vitro* process of identifying TAAs, thereby showing great potential in cancer vaccination (Immunotherapy, 2016, 8, 315). In this study, TAAs were released from the efferocytosis blockade-induced secondary necrosis. To simulate the function of the above TAAs on BMDCs, antigens in Fig. 3 were collected from the supernatants of B16F10 cells undergoing secondary necrosis. The relative explanation was added in the manuscript on page 8 lines 18-19 and page 24 lines 27-29.

6. Likewise, why are not DCs inhibited mU@OMV. Do they not express TAM/Mertk receptors?

Response: We appreciate the reviewer for these constructive questions.

PtdSer on apoptotic cells can recognize the TAM receptors, including Tyro3, Axl or MerTK, on phagocytes. Actually, besides tumor-associated macrophages, immature dendritic cells (DCs) are the other type of professional phagocytes involved in the efferocytosis of apoptotic tumor cells. However, the expression of TAM receptors is different from each other in these phagocytes. The macrophages are rich in MerTK receptor while DCs prominently express Axl (Nat. Immunol., 2014, 15, 920; Immunol. Rev., 2017, 280, 149). Therefore, high-selective MerTK inhibitor UNC2025 will not inhibit the efferocytosis function of DCs.

Although the preservation of DC efferocytosis, DCs are very limited in tumors (Nat. Biomed. Eng., 2022, 6, 44). Moreover, DCs will lose their efferocytosis capacity upon maturation (J. Exp. Med., 1998, 188, 1359), and mU@OMVs could provoke the DCs maturation, significantly weakening the efferocytosis of apoptotic tumor cells by DCs. In contrast, the tumor-infiltrating macrophages are the most abundant population in the tumor microenvironment, accounting for up to 50 % of some solid neoplasms (Nat. Immunol., 2022, 23, 1148; Cell Metab., 2019, 30, 36). Therefore, although efferocytosis can be accomplished by many types of phagocytes, macrophages are the principal phagocytes in the clearance of apoptotic tumor cells (Adv. Funct. Mater., 2021, 31, 2006220). Thus, the efficient inhibition of macrophage efferocytosis after mU@OMVs treatment led to the massive antigen release, and then potent antitumor immune responses.

7. The in vivo data in Fig. 5-8 are very nice and a strength in the paper, particularly in 2 models. Additional studies identifying mechanisms of cross presentation will provide important information to this field.

Response: Thank you for your positive comments and the kind suggestion.

In our study, antigens were first captured by mU@OMVs and then transferred to immune cell-enriched lymph nodes (LNs). CD8⁺ DCs mainly reside in the lymph nodes and are more efficient in cross-presenting tumor antigens for cytotoxic T lymphocyte activation compared with the CD8⁻ DCs (J. Exp. Med., 2009, 206, 359; Proc. Natl. Acad. Sci., 2016, 113, 1044). Therefore, we supposed that antigen-captured mU@OMVs could promote the antigen presentation of resident CD8⁺ DCs in LNs.

For verification, we comparatively investigated the antigen cross-presenting ability of CD8⁺ DCs in the draining lymph nodes (DLNs) after different treatments (Response Figure 17). Briefly, B16F10 tumor-bearing C57BL/6 mice were peritumorally injected with PBS, UNC2025, OMVs, U@OMVs, or mU@OMVs. Then, DLNs were extracted at 48 h post-administration and the expression of the major histocompatibility complex I (MHC-I) on CD11c⁺ CD8⁺ DCs was measured by flow cytometry. As expected, the MHC-I expression on CD8⁺ DCs was significantly enhanced than that in CD8⁻ DCs after mU@OMVs immunization while no obvious difference in the other groups, indicating satisfactory antigen cross-presentation in CD8⁺ DCs mediated by mU@OMVs. In summary, antigen-adsorbed mU@OMVs could facilitate the cross-presentation of tumor antigens by MHC class I molecules in

resident CD8⁺ DCs. We have revised the corresponding explanation in the revised manuscript on page 12 lines 28-30 and page 14 lines 1-4 and added these results in Fig. 5e and Supplementary Fig. 22.

REVIEWERS' COMMENTS

Reviewer #1 (Remarks to the Author):

The authors have addressed my previous concerns appropriately. The manuscript has been improved substantially. Therefore, I recommend publication of the current version in the journal.

Reviewer #2 (Remarks to the Author):

接受并发布 · 无需修改。

Editorial Note: *"Accept and publish without modification."*

Reviewer #3 (Remarks to the Author):

The authors have generally done a good job in evaluating my comments. They have clarified my points about the UNC mediated inhibition of MERTK and efferocytosis, as well as the role of TAMs in efferocytosis. I have no further queries on this paper.

Responses to the comments from the reviewers.

Reviewer #1 (Remarks to the Author):

The authors have addressed my previous concerns appropriately. The manuscript has been improved substantially.

Therefore, I recommend publication of the current version in the journal.

Response: Thanks for your recognition of our work, and we sincerely appreciate your comments, which are of great importance to improving our manuscript.

Reviewer #2 (Remarks to the Author):

接受并发布，无需修改。

Response: Thank you very much for taking the time to review our manuscript and we appreciate your kind comments.

Reviewer #3 (Remarks to the Author):

The authors have generally done a good job in evaluating my comments. They have clarified my points about the UNC mediated inhibition of Mertk and efferocytosis, as well as the role of TAMs in efferocytosis. I have no further queries on this paper.

Response: Thank the reviewer for the positive comments and appreciate the effort that you have put to review our manuscript.